

Invited perspectives. A hydrological look to precipitation intensity duration
thresholds for landslide initiation: proposing hydro-meteorological thresholds
Thom.Bogaard[1] and Roberto Greco[2]
[1] Water Resources Section, Faculty of Civil Engineering and Geosciences, Delft University of
Technology, Delft, the Netherlands: t.a.bogaard@tudelft.nl
[2] Dipartimento di Ingegneria Civile Design Edilizia e Ambiente, Università degli Studi della
Campania Luigi Vanvitelli, Aversa (CE), Italy
ABSTRACT
The vast majority of shallow landslides and debris flows are precipitation initiated. Therefore,
regional landslide hazard assessment is often based on empirically derived precipitation-
intensity-duration (PID) thresholds and landslide inventories. Generally, two features of
precipitation events are plotted and labelled with (shallow) landslide occurrence or non-
occurrence. Hereafter, a separation line or zone is drawn, mostly in logarithmic space. The
practical background of PID is that often only meteorological information is available when
analyzing (non-) occurrence of shallow landslides and, at the same time, the conceptual idea
is that precipitation information is a good proxy for both meteorological trigger and
hydrological cause. Although applied in many case studies, this approach suffers from
indistinct threshold, many false positives as well as limited physical process understanding.
Some first steps towards a more hydrologically based approach have been proposed in the
past, but these efforts received limited follow-up.
Therefore, the objective of our paper is to: a) critically analyse the concept of PID
thresholds for shallow landslides and debris flows from a hydro-meteorological point of view,
and b) propose a novel trigger-cause conceptual framework for lumped regional hydro-
meteorological hazard assessment. We will discuss this based on the published examples and
associated discussion. We discuss the PID thresholds in relation to return periods of
precipitation, soil physics and slope and catchment water balance. With this paper, we aim to
contribute to the development of a stronger conceptual model for regional landslide hazard
assessment based on physical process understanding and empirical data.





# 1 INTRODUCTION

Landsliding is the most abundant hazard having massive influence on socio-economic functioning of society. Continuous development in mountain areas increases the exposure of people and properties to the landslide hazards, with precipitation-initiated landslides being the most common. On regional scale, the probability of a landslide to occur can be assessed in different ways (Chacon et al, 2006, for review): 1) heuristic, via susceptibility modelling; 2) empirical, lumped-statistical, by relating rainfall information to the observed occurrence (e.g. Cain, 1980; Wieczorek and Glade, 2005; Guzzetti et al, 2007; Guzzetti et al, 2008, and reference therein); 3) by spatially distributed physical-deterministic modelling (e.g. Anderson and Lloyd, 1991; Montgomery and Dietrich, 1994; Wu and Sidle, 1995; Borga et al, 1998; Pack et al, 1998; Burton and Bathurst, 1998; Van Beek, 2002, Baum et al, 2008;). The heuristic models are mainly used in first assessments of hazard for regional planning. They are based on readily available static information, like topography, lithology and land use, and then empirically related to historical landslide database (if available). The dynamic predisposing factors, like actual wetness state of the potentially unstable slopes, are not taken into account. The physical process-based models can take into account the dynamics of regional hazard assessment. Most of these models run spatially distributed hydrology – slope stability calculations, with different conceptualization and degrees of complexity for the representation of the physical processes. Typically, the hydrology in these models at catchment scale is not calibrated, or the calibration is restricted to the infiltration process or local groundwater levels (if monitored). In such case, the correctness of the modelling is assessed from how well local displacements or possible failure areas can be predicted. With the increased data availability and computational power, a range of these models has been published with increased levels of complexity and applicability (e.g. Frattini et al, 2004; Arnone et al, 2011; von Ruette et al, 2013; Lepore et al, 2013; Aristizábal et al, 2016; Fan et al, 2016). However, the practical application of such deterministic models, especially in terms of early warning systems, is still limited to specific studies, due to the time effort and data demand.

The precipitation intensity-duration (ID) thresholds for hazard assessment, however, see widespread application in early warning systems, both at local and regional scale. They are based on analysis of the dynamic variables precipitation and landslide occurrence, and require a high quality spatiotemporal landslide inventory and precipitation time series. Empirical-statistical precipitation thresholds are derived by plotting two characteristics of



precipitation, usually intensity (mm/hr or mm/day) and duration (hr or days), that have or have not resulted in landslides in a given area. The separation line, a deterministic threshold or a probabilistic transition zone, between precipitation events inducing landslides and events without hazards, is then drawn visually or by separator techniques. Due to the spread of information over several orders of magnitude, it is usually plotted in bi-logarithmic scale. Various precipitation ID thresholds for landslide initiation have been derived for different physio-geographical settings and at various spatial scales (e.g. Guzzetti et al, 2007; Guzzetti et al, 2008; Wieczorek and Glade, 2005, and references therein). The global and regional landslide precipitation ID thresholds encompass different types of landslides and a distinct variety of geological and environmental factors, such as lithology, soil depths and land use. The local ID thresholds are restricted more often to relatively homogeneous conditions, and mass movement types.

However, several shortcomings are frequently recognized and discussed. For example, Berti et al (2012) recognized the problem of looking at landslide occurrence and disregarding non-occurrence when applying the ID threshold. They used a Bayesian probability approach to derive the probabilistic transition explicitly taking into account landslide occurrence and non-occurrence. Also the role of hydrology in landslide initiation, although often acknowledged to be of key importance, is usually not included in the statistical precipitation ID threshold approach. Some attempts to more explicitly include hydrology have been proposed, but however they were mainly limited to include measures for antecedent soil moisture content. However, to the authors' knowledge, these studies have never been subject to a more thorough analysis of the specific hydrological information needed for reliable local and regional hazard prediction.

Therefore, the objectives of this invited perspective are to: (a) critically analyse the precipitation ID thresholds for shallow landslides and debris flows from a hydro-meteorological point of view; and (b) propose a conceptual framework for lumped *hydro-meteorological* hazard assessment based on the concepts of trigger and cause. We will frame in this perspective some published examples and associated discussions, making reference to work made by colleagues who have already explored this avenue. Aim of this paper is to contribute to the development of a stronger conceptual model for regional landslide hazard assessment based on physical process understanding and not only on empirical data.



2 HYDROMETEOROLOGICAL ANALYSIS OF ID THRESHOLDS

COMPARING ID THRESHOLDS WITH IDF CURVES

Both precipitation intensity-duration thresholds (ID) and precipitation intensity-duration-
frequency curves (IDF) are empirical relationships linking the duration of a precipitation
event, D, with its average intensity, I=H⁄D, H being the precipitation depth fallen during the
event. IDF curves are routinely used in stormwater management design problems, as they
describe the relationship linking duration and mean intensity of precipitation events
characterized by the same return period (Chow et al., 1988). Several functional expressions
can be used to describe such a relationship (Bernard, 1932; Wenzel, 1982; Koutsoyiannis,
1998), most of which can be approximated, especially for durations longer than 1 hr, as a
power law:
$$I = A^B \qquad (1)$$

with B [-] being the slope of the log-plotted straight line and A [mm/hr] a measure of the rain
intensity of a rain event of unit duration.
Equation (1) is also adopted to describe precipitation ID thresholds, the difference
being that the IDF curves are isolines of cumulative probability of precipitation events,
whereas the ID plots are empirical thresholds for shallow landslides and debris flow
occurrence. Figure 1 gives examples of IDF curves with a return period of 10 years from
different places around the world. A common feature of the curves is that, regardless of
geographic location, B ranges from -0.8 to -0.65 for rain durations longer than ~1 hour, while
it levels off to around -0.5 for D ≤ 1 hr for most IDF curves. Note that IDF curves are mostly
determined for rain durations up to 24 hrs. In the same graph, the upper envelope of the
largest precipitation values ever observed (World Meteorological Organization, 1986), is
plotted using the equation proposed by Brutsaert (2005), which has a smaller slope with B
equal to -0.52.





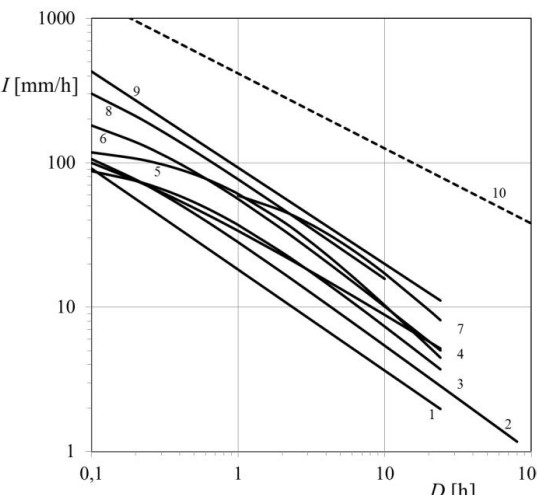


Figure 1. Examples of intensity-duration-frequency curves for 10 years return period (1-9) and curve of the maximum observed precipitations (10). Location and source: 1 Najran region, Saudi Arabia (Elsebaie, 2012); 2 Uccle, Belgium (Van de Vyver, 2015); 3 Naples, Italy (Rossi and Villani, 1993); 4 Los Angeles, California (Wenzel, 1982); 5 Pelotas, Brazil (Damé et al., 2016); 7 Hamada, Japan (Iida, 2004)); 8 Selangor, Malaysia (Chang et al., 2015); 9 Sylhet, Bangladesh (Rasel and Hossain, 2015); 10 Greatest known observed point rainfall (Brutsaert, 2005).

137

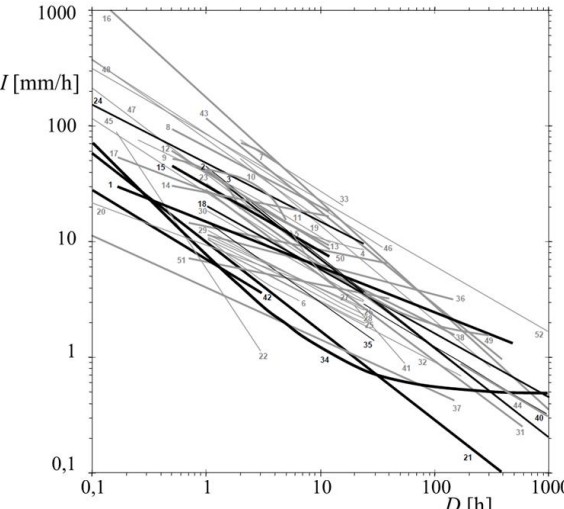

138

Figure 2. Rainfall intensity-duration (ID) thresholds. Numbers refer to case studies (Guzzetti et al, 2007). Very thick lines are global thresholds; thick lines are regional thresholds and thin lines are local thresholds. Black lines show global thresholds and thresholds determined for regions or areas pertaining to the Central to Eastern European region. Grey lines show thresholds determined for regions or areas not-pertaining to this area.



Figure 2 shows the thresholds that come from a global dataset of landslides, more than
90% of which are shallow landslides and debris flows (Guzzetti et al, 2007). Note that the
thresholds are usually obtained as lower envelope of the events resulting in landslide
initiation, although also other thresholds definitions exist (e.g. Staley et al., 2013; Peres and
Cancelliere, 2016, Ciavolella et al, 2016). Obviously, ID thresholds differ greatly between
climate and physiographic regions, especially in absolute values. Therefore, scaled
representations have been proposed for the thresholds, such as dividing precipitation intensity
by the mean annual precipitation (Guzzetti et al, 2007), in order to better compare the
thresholds. However, in our analysis the focus will be on the absolute precipitation ID
representation, as it is a convenient way to compare with IDF and for the following
discussion. The exponent of most of the reported thresholds for initiation of landslides range
between -0.2 and -0.6. By overlaying IDF and ID curves (Figure 3), for landslides triggered
by short precipitation events (D ≤ 1 hr), mostly debris flows and some shallow landslides, the
slopes of ID and IDF curves substantially coincide. On the other hand, for longer precipitation
durations, ID thresholds have smaller slopes than IDF curves. This means that landslide
initiation on the right side of the graph (lower precipitation intensity with longer duration)
would occur with rapidly increasing return periods of precipitation events. This is counter-
intuitive, as during long-lasting wet periods landslides are usually more frequent. This shows
that the method used to derive ID thresholds for landslide initiation based on landslide and
precipitation reports leads to troublesome interpretations. Several authors already pointed out
that characterizing a storm with its mean intensity, thus neglecting peaks and underestimating
actual intensity, affects the estimated probability of landslide occurrence (e.g. D'Odorico et
al., 2005; Peres and Cancelliere, 2016), and such an issue is obviously more significant for
long storm durations.



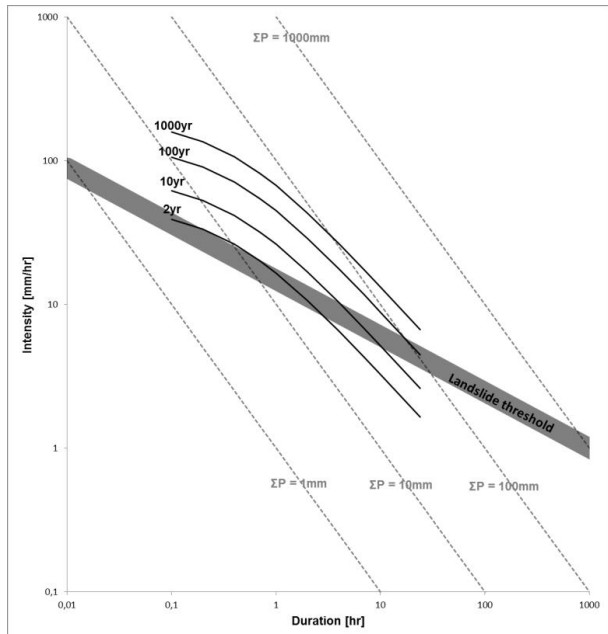



Figure 3. Schematic representation of precipitation IDF curves, isolines of accumulated precipitation and generalised ID threshold for shallow landslides and debris flows (derived from Figure 2).

HYDROLOGICAL INTERPRETATION OF ID THRESHOLDS

The precipitation ID thresholds are *"volumetric'*, i.e. they depict the total, cumulative amount of precipitation. In Figure 3 the global summary of ID thresholds for shallow landslides and debris flows (Guzzetti et al, 2007) is schematically represented by the dark grey area, but, added to it, are isolines of accumulated precipitation volume (1, 10, 100 and 1000 mm). The first observation is that the regional and global landslide thresholds clearly follow a slope different from isolines, meaning that longer duration landslide triggering thresholds require larger water volume. This is understandable if landslide size increases. However, the database consists for the overwhelming majority of shallow landslides and debris flows (Guzzetti et al, 2007). Clearly, this indicates the role of hydrology, or, to be precise, the balance between infiltration, storage and drainage capacity of a slope (Bogaard and Greco, 2015).

If we take a closer look at the volumes needed for landslide initiation, we see a spread in landslide triggers between the roughly <10 mm and >1000 mm of accumulated precipitation, with the vast majority of the empirical precipitation thresholds being reported




between 10 and 100 mm of accumulated precipitation. Under 'normal' antecedent wetness
conditions (that is, soil field capacity), an accumulated precipitation of < 10 mm is generally
not capable of triggering a landslide or debris flow. Of course, such an accumulated
precipitation volume can trigger a shallow landslide or debris flow in case of nearly saturated
antecedent conditions. In this latter case, the reported precipitation event is really the last
'push', the so-called trigger (see next section). On the other hand, precipitation volumes over
1 meter and/or durations of over 100 or even 1000 hours (> 1 month) are difficult to interpret
in terms of average precipitation intensities and triggering thresholds for shallow landslides
and debris flows. The point we want to make here is that the current ID concept combines a
too wide range of information with different types of hazards (debris flows and landslides
relate to different hydrological processes), different temporal meteorological information
(from minutes to several days). This makes the use of ID thresholds cumbersome or even
misleading.

Additionally, ID thresholds have been derived applying physically-based models of

infiltration and slope stability evaluation, which allow taking into account soil hydraulic
properties, different initial moisture conditions and the boundary conditions through which
the slope exchanges water with the surrounding hydrological system (e.g. Terlien, 1998;
Rosso et al., 2006; Salciarini et al., 2006; Frattini et al., 2009; Papa et al., 2013; Peres and
Cancelliere, 2014). Such physically-based thresholds often do not follow equation 1,
generally adopted for ID thresholds. For long precipitation durations, the physically-based ID
curves tend to flatten (e.g. Rosso et al., 2006; Salciarini et al., 2006), indicating that landslide
initiation thresholds become less sensitive to (average) precipitation intensity, which
consequently is not anymore a good explanatory variable for landslide initiation.

Interestingly, Frattini et al. (2009) followed an inverse approach and obtained

estimates of the probability of the precipitation characteristics leading to shallow landslide
initiation by considering also antecedent precipitation. In particular, they showed how the
exponent of the IDF curves of their study area (a catchment located on the east side of the
Lake Como, in Lombardy, northern Italy) changed from -0.65, for unconditional probability
of triggering events, to -0.43, when only events preceded by 300mm fallen in the antecedent
four days were considered, thus approaching the slope of the observed ID thresholds.
Antecedent precipitation can be seen as an indirect means to take into account the moisture
conditions of the soil cover before a triggering event. Therefore, the results of Frattini et al.
(2009) can also be interpreted as an indirect confirmation that considering the involved
hydrological processes would improve the performance of landslide initiation thresholds.





Greco and Bogaard (2016) give an example of the possible inclusion of slope hydrological
processes in the definition of landslide initiation threshold for the case of a slope covered by
loose granular volcanoclastic deposits laying upon a fractured limestone bedrock. The
hydraulic characteristic curves of the volcanic ashes constituting the majority of the soil cover
were known (Damiano et al., 2012; Greco et al., 2013), as well as the moisture state of the
cover before all 78 observed rainfall events (Comegna et al., 2016). Hence, it was possible to
define non-dimensional variables comparing the meteorological triggers with the infiltration
and storage capacity of the soil cover, showing that a non-dimensional hydro-meteorological
threshold performed slightly better than the precipitation ID threshold in separating events
resulting in factors of safety smaller and greater than 1.3.
3 TRIGGER - CAUSE CONCEPT: PROPOSING HYDROMETEOROLOGICAL
LANDSLIDE THRESHOLDS
In the strict sense, the precipitation ID threshold is an empirical-statistical threshold drawn to
separate failure and non-failure conditions based on observed landslides and precipitation
records. Precipitation is described in terms of intensity and duration. The main assumption is
that there is an underlying causal relation between the recorded precipitation event and the
landslide occurrence. However, by including durations up to e.g. 1 month, the direct causal
relationship is weak and the method implicitly includes the wetness state of a region. This
limitation has been recognized from the start of using ID thresholds. For several regional
hazard assessment analyses, research groups have extended the ID threshold method by
replacing the duration of a precipitation event on the x-axis with an antecedent precipitation
index (e.g. Crozier and Eyles, 1980; Glade et al, 2000). This, however, leads to limited added
information as still only precipitation information is used. On the contrary, by replacing the x-
axis with a measure for antecedent soil water content, physically relevant information is
added (e.g. Crozier and Eyles, 1980; Wilson 1989; Wilson and Wieczorek, 1995; Crozier,
1999; Glade, 2000; Chirico et al. 2000; Gabet et al, 2004; Godt et al, 2006; Ponziani et al,
2012). Interestingly, by including a water balance of the potentially unstable soil, a statistical
ID threshold evolves conceptually from a plot with one prevalent driver and data source
(precipitation) into a plot containing two predominant drivers with two distinct time-scales:
the antecedent hydrological 'cause' and the precipitation 'trigger'. Besides soil water balance
calculations, different sources of hydrological information can be used to quantify the
hydrological 'cause' of landslides. This is a largely unexplored terrain, partly as data




availability can be cumbersome and partly because physically-based, (semi-) distributed
modelling was preferred.
Concerning the 'trigger'-axis, there is little debate; it is the short-term last push for a
landslide. The time-scale for local and regional assessment of course depends on the local
situation, but hourly or daily time-scales are the most common. The 'cause'-axis should
represent the predisposing condition of an area under study. For hydrologically triggered
landslides, it should be related to the wetness state of the area. However, there are several
possible choices of hydrological variables to be plotted along the 'cause'-axis, such as soil
water content, catchment storage, representative regional groundwater level and similar.
As mentioned before, there are –besides the soil moisture storage calculations
described above- various examples of hydrological information added to landslide thresholds.
Hashino and Murota published in 1971 an analysis of landslide triggers in a catchment,
related to debris production, using measured river discharge data to link it to the water
balance of the catchment. They identified that the landslides in their study area took place in
antecedent conditions of catchment storage above average. This is one of the earliest reported
studies we know of explicitly looking at catchment water balance as an important source of
information on the antecedent hydrological condition of an area in relation to landslide
occurrence. Reichenbach et al (1998) made a combined flood and landslide hazard analysis of
the Tiber river catchment using 72 years of historical daily discharge data from different
gauging stations where hydrological parameters were calculated, such as maximum mean
daily discharge, discharge intensity and flood volume and duration. Probability of occurrence
of landslides and floods was based on the ranking of the events. Combining maximum mean
daily discharge and discharge intensity, regional hydrological thresholds for landslide and
flood hazard (individually or combined) could be drawn. Chitu et al (2016) followed a
somewhat similar approach, analyzing the river discharge in several catchments in the
Ialomita Subcarpathians in Romania for landslide events occurred in 2014. The catchments
could be characterized as having low/high storage. Additionally, a calibrated regional rainfall-
runoff model was used for hydrological analysis of landslides in particular catchments.
Detailed analysis of the (modelled) hydrological response indicated that in two catchments
with low permeability the direct runoff was strongly related to landslide occurrence, whereas
it was linked to modeled soil infiltration flux in the more permeable catchment. In some cases,
regional groundwater level could be informative. Bogaard et al (2013) studied the hydro-
meteorological triggering threshold of the re-activating coastal Villerville–Cricqueboeuf
landslide, Normandy, France. In this situation the hinterland of the coastal cliff consists of a



well-defined regional groundwater level. Landslide reactivation was seen to take place only
when water level was in the upper, more permeable top layer. The triggering rain event
together with surpassing a certain regional groundwater threshold could explain 3 of the 4 re-
activations. Note that these groundwater levels were not taken in the active landslide area but
several kilometers inland.
Recently, further attempts have been made to use river discharge and lumped water
storage in a catchment as a proxy for the predisposing conditions for landslides along its
slopes. Following Hashino and Murota (1971), the basic idea is that when 'more-than-
average' water is stored in the catchment, it is more likely that a rainfall event triggers
landslides. The disadvantage of using catchment wide storage is the relatively low spatial
resolution, and the difficulty of having (reliable and homogeneous) discharge time series in
catchments. Moreover, catchment storage assessment needs information on evaporation which
can have significant uncertainties. Of course, such an approach works only if the causes of the
predisposing conditions for landslides are somewhat related to catchment scale hydrological
processes. Ciavolella et al. (2016) defined a cause-trigger hydro-meteorological threshold in
the catchment of river Scoltenna, in Emilia Romagna (Italy), linking catchment storage and
event rainfall intensity, and compared its performance with that of a statistical ID
precipitation threshold. The two thresholds performed similarly, with the hydro-
meteorological thresholds resulting more accurate in the identification of landslides, but
giving a somewhat larger number of false positives.
The above examples indicate that considering hydrological causes could be useful for
a better identification of landslide initiation, but, at the same time, they show that the correct
identification of the hydrological processes involved in the establishment of the predisposing
conditions for landslides in a considered area is mandatory for choosing the most informative
hydrological variable to be plotted along the x-'cause'-axis.

4 CONCLUDING REMARKS AND OUTLOOK

The intrinsic limitation of precipitation ID thresholds for the identification of landslide
initiation conditions has been pointed out long since. Indeed, such thresholds neglect the role
of the hydrological processes occurring along slopes, which lead to the predisposing
conditions for failure (causes), and focus predominantly on the characteristics of the last
rainfall event leading to slope failure (triggers). As a consequence, the predictive value of the
ID thresholds is often low, even when they refer to small areas. We argue that the threshold



values for rainfall intensity of short and long duration (the far left and right side of the graphs)
have limited physical meaning and, consequently, that the use of precipitation ID thresholds
can lead to misleading interpretation of initiation conditions, as important antecedent
conditions and rainfall intensity variations are not taken into account. For this reason, we here
advocate to be very careful in uncritically using the precipitation ID thresholds as kind of
regional characteristic of (shallow) landslide occurrence.

Equally, for this and several other reasons, many colleagues advocate the use of
spatially-distributed physically-based models for assessing landslide probability. The obvious
downside is that large data input and a well calibrated model are required. However, it is fair
to say, data are becoming more and more available and even precipitation predictions are
improving rapidly, especially with short lead-time. The use of high quality rainfall prediction
with very short lead time (e.g. 3 hours) requires efficient numerical models combined with
high computational power, especially if predictions are used for early warning purposes. This,
in practice, is still easier said than done. Therefore, we believe, that lumped, empirical (or
semi-empirical) thresholds will continue having a practical value, which still justifies
scientific attention.

We propose to use the cause-trigger concept for defining regional landslide initiation
thresholds. This, we agree, is challenging, but, in our opinion, not impossible. Looking at the
discussed examples, it becomes clear that the choice of the most informative hydrological
variable to be used as a proxy for landslide predisposing conditions strictly depends on site-
specific geomorphological characteristics, and that accurate analysis of the boundaries
through which the potentially unstable area exchanges water with the surrounding
hydrological systems is mandatory. In other words, for the assessment of landslide
predisposing conditions, the water balance of the slope should be assessed, but getting the
information about the involved hydrological processes (e.g. evaporation, runoff, groundwater
recharge) at the required spatial-temporal resolution is often still a challenge, although remote
sensing could help increasing the reliability of catchment water storage estimates.







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
