# Peer review of "Hydrological perspectives on precipitation intensity duration thresholds for"

_Natural Hazards and Earth System Sciences, 2017_

## Referee Comment (RC1) · Anonymous Referee #1 · 17 Aug 2017

**General comments**

The paper offers a hydrological perspective of precipitation intensity-duration thresholds (hereafter, ID thresholds) for landslide triggering, useful in early warning systems. The ID threshold is a well established empirical model, as it is proposed in numerous studies. Several limitations affect these thresholds, as summarized in this paper. The authors with this paper propose to move away from this "conventional" path for future research, arguing that simple, even lumped, hydrological information should be introduced. They propose a general framework, where thresholds should represent both

landslide causes (dynamic predisposing conditions) and landslide triggers. They argue that with ID thresholds only the latter are (conceptually) considered. Hydrological information is related to the former, and should be represented by something linked to soil water content.

I overall think that this is a good paper and well written. On the other hand, I also think that some improvements can be made. In particular, two main issues the authors should better discuss are:

1. How to separate between landslide "causes" and landslide "triggers" in practice? In other words: at which instant/timescale one should think that there is a switch from causes to triggers?

2. How to manage the higher modeling freedom (respect to PID thresholds) that one can introduce by hydrological analyses?

More details on these two points are given in the specific comments (comments to L 253 and L 262-264).

Finally, I recommend minor revisions for this manuscript.

**Specific comments**

L 20: "the conceptual idea is that precipitation information is a good proxy for both meteorological trigger and hydrological cause". It cannot be said that, in general, researchers deriving ID thresholds and their users have this conceptual idea in mind. This is a move of the authors which is not fully justified. So I think that this sentence should be rewritten, perhaps writing something on the fact that it is in general thought that precipitation information can be linked by simple relationships to landslide occurrence, without explicitly taking into account hydrology.

L 22: It is not fully clear what does "indistinct threshold" mean

L 36: "landslide is the most abundant hazard". Are the authors sure that "landsliding

is the *most abundant* hazard"? Maybe say that it is "*one of the most* abundant *natural* hazards", and add some references to literature (for instance: Sidle and Ochiai, 2013)

*Sidle, R. C. and Ochiai, H.: Landslides: Processes, Prediction, and Land Use, Water Resources Monograph, 2013.*

L 39 – 45: The three approaches listed by the authors are not all aimed to assess "landslide probability" in a strict sense (only number 3 is). In fact approach (1) leads to an assessment of landslide "susceptibility", which is not exactly a probability, but an index of landslide proneness in a relative scale. Approach (2) does not provide in general landslide probability, as most of the landslide triggering threshold schemes are "deterministic", and probability is in fact only in theory – but very seldom in practice – related to landslide triggering thresholds (Aleotti, 2004; Iiritano et al., 1998). The authors should clarify this point.

*Aleotti, P.: A warning system for rainfall-induced shallow failures, Eng. Geol., 73, 247– 265, 2004.*

*Iiritano, G., Versace, P., Sirangelo, B., 1998. Real time estimation of hazard for land- slides triggered by rainfall. Environmental Geology 35 (2– 3), 175– 183.*

L 42: perhaps integrate literature on this, with other more recent papers (e.g. Peruc- cacci et al., 2017 and references therein)

*Peruccacci, S., Brunetti, M. T., Gariano, S. L., Melillo, M., Rossi, M. and Guzzetti, F.: rainfall thresholds for possible landslide occurrence in Italy, Geomorphology, 290, 39–57, doi:10.1016/j.geomorph.2017.03.031, 2017.*

L 46: The term *hazard* may have a specific definition in the natural hazards field, re- lated to the *probability of the event to occur*. So the authors should clarify that they refer to "hazard" in a broader sense. Perhaps in clarifying this they should cite a generally accepted definition of "landslide hazard". This comment is related to preceding one on L 39 - 45

[Figure]

L 63: the authors use both ID / PID when referring to precipitation intensity and duration thresholds. Only one way should be used

L 63: "hazard" is perhaps not fully appropriate

L 70: add references to papers where a "probabilistic transition zone" is used

L 88: It seems that authors are referring to works where antecedent precipitation is used (perhaps as a "measure of antecedent soil moisture content"). Here the authors should better clarify what they are referring to, and cite pertaining papers.

L 88: It is unclear if antecedent precipitation should be seen in the authors' framework as an hydrological (cause) or meteorological (trigger) variable

L 90: again, here "hazard" is perhaps not fully appropriate

Figures 1 to 3: perhaps for a better comparison of the various curves it may be useful to plot in planes with the same axis range (e.g. x-axis of Fig. 1 goes from 0.1 to 100, while Fig. 2 from 0.1 to 1000). Also, it may be better that figures have the same appearance (e.g. no grid in the plot of Fig. 1; adjust font size in Fig. 3).

Figure 3: It is unclear how the dark grey area representing "landslide threshold" is derived from figure 2, as the area that it covers is narrower than that covered by thresholds in Fig. 2

L 171: It is unclear in which sense the ID threshold is "generalized"

Figure 3: P is undefined (though its meaning can be easily understood from discussion in the text).

L 175: It is not clear why precipitation ID thresholds are "volumetric", as an infinite number of (I,D) or (H,D) pairs can be associated to a given event rainfall H.

L 181: It is unclear why greater precipitation volumes should imply bigger landslides. Is this something reported in literature? I imagine that this is in general not true, as

the amount of rainfall gives little (or none) information on its spatial extension, and thus of that of the landslide. Also ID thresholds are derived using databases that usually report little information on landslide size, and to say that "the database consists for the overwhelming majority of shallow landslides and debris flows" doesn't mean that the size of landslides is small.

L 192 – 196: In this discussion the authors should mention that ID thresholds are sensitive to the way a rainfall event is defined, that is, mainly the maximum zero-precipitation interval within a rainfall event (See Vessia et al., 2014; Melillo et al; 2015). Cleary, the shorter this interval is, the shorter the length of rainfall events will be. With long maximum dryness the events can be so long that different hydrological processes can take place. In this case rainfall events do not represent "the last push" but a mixture between "causes" and "triggers".

*Vessia, G., Parise, M., Brunetti, M. T., Peruccacci, S., Rossi, M., Vennari, C. and Guzzetti, F.: Automated reconstruction of rainfall events responsible for shallow landslides, Nat. Hazards Earth Syst. Sci., 14(9), 2399–2408, doi:10.5194/nhess-14-2399-2014, 2014.*

*Melillo, M., Brunetti, M. T., Peruccacci, S., Gariano, S. L. and Guzzetti, F.: An algorithm for the objective reconstruction of rainfall events responsible for landslides, Landslides, 12(2), 311–320, doi:10.1007/s10346-014-0471-3, 2015.*

L 253: The authors should discuss how to separate between the time scales of "causes" and those of the "triggers". In other words, how to switch, in practice, from the "cause" hydrological analysis (storage), to the "triggers" meteorological analysis (rainfall)? In other words, how does the framework the authors propose contribute in removing the subjectivity of identifying the rainfall that represents the "trigger"/"last push" (see comment on L 192 – 196)?

L 253: Another point is: hydrology may be in general important also during the "triggering" process, while in the authors' framework it is not explicitly taken into account. Are

the authors implicitly saying that "hydrology of the last push" can be taken into account without a significant processing of rainfall data?

L 262-264: "However there are several possible choices of hydrological variables to be plotted along the cause-axis, such as soil water content, catchment storage, representative regional groundwater level and similar". This implicitly reveals that a high degree of subjectivity follows from the framework that the authors propose. Researchers do generally agree that subjectivity of the ID threshold assessment is significant, in spite of his simplicity. For instance, one source of subjectivity in ID thresholds is related to the choice of the maximum zero-precipitation interval to define rainfall events (see comment on L 192-196). This is known to impair comparisons between thresholds, which thus makes it difficult to search for general landslide triggering thresholds. The framework that the authors propose seems to possibly bring a higher heterogeneity of the analyses, and thus maybe can in practice represent a step backwards for finding unifying concepts. By introducing hydrological analysis, researchers may have more freedom in choosing models and parameters for estimating the "cause" variable (antecedent soil water content). This may represent a possible way to manipulate the results so that the performances of the resulting hydro-meteorological thresholds appear to be higher than they actually are. Thus, the authors should discuss how one can prevent this, perhaps by highlighting the importance of always performing validation analyses, i.e. to test developed thresholds against a sub-dataset which is not used in calibration.

L 319: "ID thresholds neglect the role of the hydrological processes" is a strong statement. Indeed it may be written that hydrological processes are too simplistically represented by ID thresholds. In other words, precipitation is the main cause of landslides, but the main problems is: how to process precipitation information to obtain thresholds that perform well in forecasting landslides? And, of course, ID thresholds certainly do not represent the best way to processes rainfall data.

L 332: I agree that one downside of spatially-distributed physically based models is that

they require a "well calibration". However to estimate catchment storage (as in Ciav-olella et al., 2016), requires a well calibrated model too. The authors should discuss better this point.

L 233: A sketch explaining the approach the authors propose can be useful for readers.

**Technical corrections**

L 42: Caine instead of Cain

L 45: maybe something is missing as citations finish with a ";"

L 71: "separation" instead of "separator"

L 78: remove "," after "conditions"

L 142: perhaps replace "for regions or areas not pertaining to this area" with "other regions or areas"

L 147: "threshold" instead of "thresholds"

L 197: perhaps "phenomena" instead of "hazards"

L 198: "related" instead of "relate"

L 223: "thresholds" instead of "threshold"

L 255: perhaps "field" instead of "terrain"

L 283: "specific" instead of "particular"

L 318: "limitations" instead of "limitation"

L 326: "interpretations" instead of "interpretation"

L 331: "physically based" instead of "physically-based"

---

## Referee Comment (RC2) · R. C. Sidle (Referee) · 18 Aug 2017

This paper offers a refreshing and needed critique of the ID relationships commonly used in regional (and global) landslide predictions. Furthermore, it proposes an improved approach that includes metrics of predisposing 'causes' and 'triggers' of shallow landslides. As such, it should stimulate new research in this arena that will benefit landslide and hillslope debris flow prediction. It will be a valuable contribution to NHESS with some moderate revision.

[Figure]

I noted several times in my review that follows that ID relations (at least some previously published ones) have erroneously reported data as 'individual storms', which were obviously not individual events (i.e., very, very long durations). Additionally, on the opposite side of the ID 'x-axis' there are instances of very, very short storms of high intensity triggering landslides – these appear to be bursts of intensity on saturated soils as noted by the authors or they could in fact represent a totally different process, like channel bed mobilization causing a debris flow. My recollection of reading through some older reports in which data were used to develop ID thresholds is that in some cases the described mass failure was more of a within channel debris flow. Off the top of my head, I am thinking of some of Rapp's early papers that were included in Caine's threshold. In any event, these anomalies should be considered or mentioned herein.

I have noted a number of editorial suggestions directly on the manuscript which I will attach for the authors. More scientific technical comments are noted as follows:

Title: I would say "Hydrological perspectives on precipitation intensity – duration thresholds….."

Lines 28-29: reword – "discuss" based on "associated discussion"

Lines 86-88: yes, we tried this in our 1985 Hillslope Stability and Land Use book using antecedent rainfall information, but the problem was the lack of documentation of such antecedent rainfall data in earlier studies. Overall, we felt that it did improve the ID thresholds (at least conceptually).

Line 109: The term 'stormwater management' implies to me more of an urban planning context; that may be my bias, but you may want to add 'flood prediction' (or something like this) as well.

Line 125-126: I think that this is a key difference between practical applications of IDF and ID curves; that is most (or at least many) shallow landslides respond to sort-term intensity bursts which are not articulated in typical IDF's. You may want to mention this.

Line 144: Try not to start sentences with "Figure x shows..."; this can be seen in the Figure and caption. Just directly say what you wish to say about the data in the figure and cite the figure in parenthesis at the end of the sentence.

Lines 155-157: rework this sentence – understandable, but a bit confusing. Maybe just put 'mostly debris flow and some shallow landslides' in parenthesis. Furthermore, I think there are some issues with such very short 'landslide producing storms' reported in the literature that are captured in these cited thresholds. As you note, they are probably mostly debris flows, and upon inspection of some earlier papers that reported such short-term events, it seemed that the authors were referring to possibly a different process – e.g., debris flows caused by channel bed mobilisation. I looking into this matter in our 2006 landslide book, we actually threw out some of these short-term rain events when constructing new ID curves because we were convinced that they represented different triggering processes.

Lines 158-166: I agree that this is problematic, and I feel (as you state) that ignoring short-term peaks of rainfall in an otherwise long-duration, lower intensity event is the main reason for this problem. Based on my work and that of others, I always say that one common scenario for shallow rapid landslide initiation is a long storm of low to moderate intensity, with a peak intensity occurring near the end of the event. Another issue here, I agree that the longer return periods for landslides triggered by long-duration, low intensity storms is counterintuitive; however, when we looked into the actual data for some of these so-called long duration events that triggered landslides (in reviewing references for the 2006 AGU landslide book), it became apparent that some of the data included in these ID relationships were not strictly 'individual events', rather these were based on a longer period of rainfall leading up to the landslide. – thus, a direct comparison with some of these so-called long-duration landslide triggering 'events' with IDF curves for actual individual events may be a bit problematic. You probably should mention this potential discrepancy. My point is probably only relevant for the very long 'events', but it may be worth mentioning.
Lines 185-188: This sentence is a bit confusing; it seems that you are referring to reported data when you saw the 'vast majority of empirical thresholds fall between . . .". Are you saying that for other studies most of the landslide reported would fall between thresholds of 10 to 100 mm? If so, you need to cite some references. But I am not sure that is what you are trying to say here. Anyway, please clarify. (and you overuse the expression 'vast majority' – just say most or the majority).

Lines 194-196 See my previous comment about data for very long duration 'events' that are likely not individual events.

Lines 209-210: In addition to my comment in the text, also see my previous comment about data for very long duration 'events' that are likely not individual events.

Lines 227-231: Very complex sentence and a bit awkward. Can you rewrite this or try to break it up a bit?

Lines 240-241: I don't mean to be beating a 'dead horse' again, but such long events are obviously not 'events'; they were probably included in databases because this was the only precipitation record reported.

Line 257: Why do you say 'was preferred"? by who?

Lines 258-260: Reword the first sentence to note that the trigger axis refers to the rainfall characteristics (intensity) responsible for initiating the landslide. When you say "depends on the local situation" – I think you mean both available data and the rainfall characteristics that are responsible for landslide initiation in that area.

Line 276: What do you mean by 'discharge intensity"? This is a rather unconventional term.

Line 282: What is low/high storage?

Lines 286-287 (and the sentences that follow): I think this phenomena occurs for deep-seated landslides like earthflows of slump-earthflows – maybe better to state this to

avoid confusion, because you are mostly focussing on shallow landslides. There is some older work in Japan that has clearly showed such relationships with earthflow reactivation and a threshold groundwater depth. I believe mark Reid also published a paper on this from earlier work in Hawaii.

Line 322: Again, not all data in these ID relationships were for individual events.

Line 323: you mean even when they are developed for small areas?

Line 338: These will be particularly valuable in developing countries.

I really like the message in the last paragraph of the Conclusions! Well articulated.

Roy Sidle

---

## Short Comment (SC1) · 4 Sep 2017

The authors analyze the concept of precipitation intensity-duration (ID) thresholds for shallow landslides and debris flows from a hydro-meteorological perspective to propose a new approach to the problem. The contribution is largely welcome since it suggests new approaches and perspectives to overcome important, but sometimes neglected, limitations of the commonly used methods.

Within the interesting analysis of the relationship between ID thresholds and IDF curves, the discussion so far neglects the impact of rainfall estimation uncertainty and its possible dependence on rainfall duration and return period (e.g., Krajewski et al., 2003, www.dx.doi.org/10.1623/hysj.48.2.151.44694; Ciach and Krajewski, 2006, www.dx.doi.org/10.1016/j.advwatres.2005.11.003).

Recently, rainfall estimation uncertainty caused by the use of rain gauge measurements (still the most common source of rainfall estimates in this field) was shown to significantly affect the derived ID thresholds causing systematic bias (Nikolopoulos et al., 2014, www.dx.doi.org/10.1016/j.geomorph.2014.06.015). This systematic bias is caused by (i) systematic rainfall patterns observed around the triggering locations and (ii) the use of log-transformations within the derivation of the ID (Marra et al., 2016, www.dx.doi.org/10.1016/j.jhydrol.2015.10.010). At least for durations

This was a technical comment on an introductory aspect of the study; this being said, I repeat my compliments to the authors for the manuscript and the new proposed perspective.

With kind regards, Francesco Marra

---

## Short Comment (SC2) · 6 Sep 2017

**B. Mirus**

bbmirus@usgs.gov

Received and published: 6 September 2017

The authors present a much needed discussion about some systematic problems with precipitation intensity-duration (ID) threshold approaches for predicting shallow landslides. In particular, they point out an unfortunate lack of reasonable constraints on the max/min duration of rainfall events, and also discuss how these unbounded events can affect the average intensity and predictive capabilities. Both issues are largely ignored in many studies focused on developing and testing ID thresholds, so it's a worthy discussion about some crucial sources of error. The authors also highlight the potential

importance of hydrological information in addition to precipitation characteristics, which have not been systematically incorporated into landslide early warning criteria. In my opinion the most innovative contribution presented in the manuscript is the comparison of the rainfall intensity recurrence intervals and ID thresholds for landslide initiation from the literature, along with the contour lines of cumulative storm totals (Fig. 3). This is a new and intuitive way to broadly illustrate their point about some problems with the ID threshold concept. However, my primary concerns with the manuscript are twofold:

(1) Limited concrete guidance is provided on how to apply the proposed "cause-trigger" framework, so the potential novelty of the approach seems somewhat overstated.

The general concept that both the predisposing factors (e.g. antecedent wetness) and a rainfall triggering event are needed to explain shallow landslide initiation is already generally accepted and has in fact been implemented in a number of landslide initiation thresholds. For example, two different rainfall thresholds developed for the Seattle area explicitly account for antecedent factors: (a) the recent-antecedent cumulative precipitation threshold compares the 3-day triggering rainfall to the 15-day antecedent rainfall (Chleborad et al, 2008), and (b) the Antecedent Water Index is used with an exponential ID threshold for events between 10min and 10days in duration, though storms are generally less than 24hours (Godt et al., 2006). The authors also cite several other papers (including some of their own published work) that in various ways incorporate antecedent wetness as a measure for the predisposing factors prior to the triggering rainfall event using soil water balance modeling or catchment storage. As such it is not clear how the "cause-trigger" approach is truly novel, but rather seems to be a new term for a topic in need of further exploration. (As an aside, the term "predisposing factors" seems to be a more appropriate term... without context "cause" could be misleading since both predisposing factors and a triggering event are needed to cause a landslide.)

(2) The critical issue of data availability is understated.
The topic of data availability is largely avoided until the very end of the conclusions, at which point it comes across as an afterthought instead of the main reason the precipitation ID threshold has been employed successfully for decades. Without continuous records of appropriate data during historic landsliding events it is challenging (if not impossible) to develop and test alternatives to the precipitation ID threshold. The reality is that rainfall data is widely available and has been for some time, which has facilitated useful, albeit somewhat flawed tools for assessing landslide potential for a number of landslide-prone areas. Secondly, rainfall can be predicted in advance with considerable accuracy, so despite some errors in ID thresholds, the trade-off between appropriate lead-time using weather forecasts and threshold accuracy must at least be considered when arguing for alternative threshold approaches. Without a more balanced discussion of data availability it's not entirely clear whether the authors are arguing for better analysis of rainfall data that distinguished between the "causing" rainfall and the "triggering" rainfall or if the authors suggest that rainfall is not an appropriate data source for the "cause" variable and the ID threshold concept has been employed incorrectly for very long and very short duration storms. Although the Invited Perspective highlights both these problems with the ID threshold approach, it remains unclear how the "cause-trigger" approach can be used to solve these problems within the context of limited data availability.

After addressing these two issues regarding the novelty of the proposed approach and the availability of data for landslide initiation thresholds, the authors' hydrologic perspectives on the precipitation ID approach for landslide prediction will be a valuable contribution and will surely be of considerable interest to readers of NHESS. I suggest a number of general and specific revisions prior to publication, outlined in the sections below.

Thank you for your consideration.

Kind regards, Ben Mirus
General Revisions:

(a) Explain how the details of how the "cause-trigger" concept can be distinguished from prior contributions that consider antecedent conditions, or qualify the novelty of the proposed approach within the context of such prior work.

(b) Provide more concrete guidance on how the "cause-trigger" framework could be applied for future studies. In particular, how should researchers constrain the duration of storms to distinguish between "cause/predisposing factors" and "trigger"?

(c) Include a more balanced discussion of what data could reasonably be obtained to inform the "cause" axis for any landslide early-warning threshold relative to the widely available (and forecastable) input of rainfall.

Specific Edits:

L1: I agree with the revision to the title suggested by Roy Sidle and would further suggest removing the second phrase since there no specific hydro-meteorological thresholds are proposed. Thus the suggested title is shorter and more precise: "Hydrological Perspectives on the Precipitation Intensity Duration Thresholds for Shallow Landslide Initiation"

L13: Provide some citation or definitive evidence for the strong, yet disputable statements like "vast majority" and "never" ... otherwise such pronouncements should be avoided in scientific writing. Furthermore, as is later argued in the manuscript, precipitation does not actually initiate the landslide. Suggest revising to: "Many shallow landslides and debris flows are rainfall induced."

L22: What does "indistinct" mean? Thresholds are by nature distinct. On the other hand, the errors resulting from application of distinct thresholds over broad areas reflects the heterogeneity of natural systems. In theory, each hillslope/hollow has a unique threshold that must be averaged over some area and some time to create a useful tool for landslide early warning.
L27: Again, calling this a novel conceptual framework is an overstatement. See general comments.

L36: References to support this claim? I was not aware landsliding is the MOST abundant hazard. At the very least it should be qualified as a natural hazard, since many health or other hazards could be considered more abundant and/or detrimental so socio-economics.

L40-43: Unclear from the description provided here how 1) and 3) are different when applied to assessing landslide probability. Perhaps some example citations later in the paragraph could help distinguish between the two.

L59: I recommend also citing Anagnostopoulos et al. 2015 when discussing model complexity.

L74-75: Perhaps include some more recent citations that are less than 10 years old?

L78-79: Yes. Also there is considerable error introduced by the heterogeneity that must be "averaged out" for a PID threshold to be developed over an area of interest.

L86-88: Are you proposing something that is better than soil moisture? It seems that soil moisture would be better than the other variables suggested (albeit harder to measure), so it is seems counterintuitive to state that these studies are "limited" to measures of antecedent moisture content.

L88: Never say never. In general it is unwise use this word in scientific writing unless it can be rigorously confirmed, which is almost "never" possible. Suggest revising to "not" or "have not been the subject of"

L106: Here and elsewhere the abbreviation switches from PID to ID... either is fine, but use only one consistently throughout.

L149: What is an "absolute" value of a threshold? Do you mean, for example the xand y-intercept values? Revise for clarity. Interactive comment
L152: Again, what is the "absolute" precipitation ID?

L153: At some point in this part of the discussion you should mention the novel use of duration-frequency curves by Fusco et al., 2017. They use this concept to examine temporal and spatial patterns of pressure head states that predispose slopes to recharge and/or landslide initiation. I think it is a nice example of how the "cause" concept you propose could be implemented practically.

L167: Yes. This also leads to questions about how storm durations are defined, particularly since longer storms are more likely to include actual breaks in precipitation where drainage and ET can be more effective in reducing landslide initiation potential.

L181-184: Revise these sentences for greater clarity. It seems like the main point is that if larger cumulative precipitations are needed to initiate landslides at lower intensities we would expect that to be reflected by the larger landslides in the inventory, but the inventory you reference is mostly small landslides, so an alternate interpretation is that the slope drains while it's raining? At least the last sentence is incomplete to communicate the message more clearly.

L186: Technically this (between <10 and >1000) is not a range, it is unbounded. Do you mean between >10 and
L206: Napolitano et al., 2015 is another good reference to include here as they also used seasonal variations in antecedent soil wetness to identify different thresholds for winter vs. summer.

L218-219: Indeed, this is the concept underlying the recent-antecedent cumulative rainfall threshold of Chleborad et al., 2008, except they use prescribed durations, which have since been statistically tested with receiver operator characteristics (Scheevel et al., 2017)

L229-231: The wording of this sentence is confusing. What is the significance of this separation between near-failure events at FS

groundwater recharge and no runoff (or landslides?). Perhaps more relevant would be the thickness and hydraulic conductivity of the soil, which ultimately are reflected by how quickly the catchment drains.

L290-291: There are a lot of things mixed up in here, which makes it difficult to relate to the primary topics of the article. First, this is not a shallow landslide, so perhaps this is a bit of a tangential argument for this paper, but it seems that the main point is mobility for a large, slow moving landslide can be related to groundwater levels. That's fine. However, it's not clear groundwater levels would be a good proxy for conditions favoring shallow landslides, particularly since deeper groundwater levels might not respond until after shallow soils on hillslopes have drained and are no longer susceptible to failure.

L310-314: OK. This makes sense, but can you provide more concrete guidance or framework for evaluating the appropriate "cause" variable? Also, can you provide some balanced perspective of how readily available those types of data may be relative to rainfall? An example Figure 4 might be helpful.

L322-323: This is a somewhat subjective (i.e. value) judgment, which is tangential to the discussion presented here. The perceived or tangible value of predicting even 1/100 landslide events correctly at the expense of many false alarms is an entirely different question. Probably "predictive accuracy" would be more appropriate.

L340-349: I completely agree with these statements and don't wish to argue with the sentiment, but at the same time the conclusions are rather wordy and not particularly satisfying or informative. Another (shorter) way of saying this is that hydrologic information could improve individual thresholds for shallow landslide initiation, but the type of hydrologic information that is most appropriate will vary based on location and data availability. So then how do we go about addressing this issue?

L350: The last-minute mention of remote sensing comes across as a bit of an afterthought and it is not clear how this very broad suite of information products can be used to constrain hillslope water balance. Why not soil moisture monitoring? Interactive comment
**References Cited:**

Anagnostopoulos, G.G., S. Fatichi, P. Burlando, 2015, An advanced process-based distributed model for the investigation of rainfall-induced landslides: The effect of process representation and boundary conditions, Water Resources Research, doi:10.1002/2015WR016909

Chleborad, A.F., R.L. Baum, J.W. Godt, P.S. Powers, 2008, A prototype for forecasting landslides in the Seattle, Washington, Area, Reviews in Engineering Geology, doi: 10.1130/2008.4020(06).

Fusco, F., V. Allocca, and P. De Vita, 2017, Hydro-geomorphological modelling of ashfall pyroclastic soils for debris flow initiation and groundwater recharge in Campania (southern Italy), Catena, doi:10.1016/j.catena.2017.07.010.

Godt, J.W., R. L. Baum, A.F. Chleborad, 2006, Rainfall characteristics for shallow landsliding in Seattle, Washington, USA, Earth Surface Processes and Landforms, doi: 10.1002/esp.1237.

Napolitano, E., F. Fusco, R. L. Baum, J. W. Godt, P. De Vita. 2015, Effect of antecedenthydrological conditions on rainfall triggering of debris flows in ash-fall pyroclastic mantled slopes of Campania (southern Italy), Landslides, doi:10.1007/s10346-015-0647-5)

Scheevel, C.R., R.L. Baum, B.B. Mirus, J.B. Smith, 2017, Precipitation thresholds for landslide occurrence near Seattle, Mukilteo, and Everett, Washington: U.S. Geological Survey Open-File Report 2017–1039, doi:10.3133/ofr20171039.

---

## Author Comment (AC1) · 23 Oct 2017

**Nat. Hazards Earth Syst. Sci. Discuss., https://doi.org/10.5194/nhess-2017-241**
Invited perspectives. A hydrological look to precipitation intensity duration thresholds for landslide initiation: proposing hydro-meteorological thresholds
**Reply to Referee # 1**

We thank the reviewer for the detailed and constructive review. Below we address and reply to the comments and questions. In Italic typesetting the original review is given, and in roman typesetting our replies.

Thom Bogaard and Roberto Greco

**General comments**
*The paper offers a hydrological perspective of precipitation intensity-duration thresholds (hereafter, ID thresholds) for landslide triggering, useful in early warning systems. The ID threshold is a well established empirical model, as it is proposed in numerous studies. Several limitations affect these thresholds, as summarized in this paper. The authors with this paper propose to move away from this "conventional" path for future research, arguing that simple, even lumped, hydrological information should be introduced. They propose a general framework, where thresholds should represent both landslide causes (dynamic predisposing conditions) and landslide triggers. They argue that with ID thresholds only the latter are (conceptually) considered. Hydrological information is related to the former, and should be represented by something linked to soil water content.*
*I overall think that this is a good paper and well written. On the other hand, I also think that some improvements can be made.*

We thank the reviewer for the detailed and constructive review. Actually, our conceptual approach is more general than only focusing on shallow landslide (then, it would be indeed soil moisture). Our point is that different hydrological information could be useful for landslide hazard assessment.

*In particular, two main issues the authors should better discuss are:*
*1. How to separate between landslide "causes" and landslide "triggers" in practice?*
*In other words: at which instant/timescale one should think that there is a switch fromcauses to triggers?*

This is an excellent point that we discussed at length ourselves. The reviewer is correct that categorizing landslides based on "cause" and "trigger" requires a kind of time scale to separate the two. The discussion on the timescale of trigger-cause is already half a century old (e.g. Sowers and Sowers, 1970). Wieczorek (1996) defined triggering as an "external stimulus (…) that causes a near-immediate response in the form of a landslide by rapidly increasing the stresses or by reducing the strength of slope materials.". In our own advanced review WIREs Water (Bogaard-Greco, 2015) we summarized the trigger-cause as: "A trigger is thelast push for a slope to become unstable, whereas thecause is the underlying, often long term, change thatoccurred preparing the slope for failing.".
So we see the trigger as 'the last push' with near-immediate effect and consequently the hydrological cause is all before that. We agree that when adapting our proposed framework, to use the cause-trigger concept for defining regional landslide initiation thresholds, it becomes eminent to start defining timescale to distinguish between trigger and cause.This will be different for different landslides and slopes.
We agree with the reviewer we did not discuss this and we will add a short discussion on the definition of the timescale of the trigger event both in introduction and in conclusion section, as well as in the description of the discussed examples, which point out how the timescales of both trigger and cause are strongly related to the effective hydrological processes of a specific site.. However, for us, the more 'mathematical' or 'precise' defining of the trigger timescale in the various situations is out scope of this invited perspective and more for follow up work, when the community starts adapting the proposed concept.

*2. How to manage the higher modeling freedom (respect to PID thresholds) that one can introduce by hydrological analyses?*
The reviewer is correct that looking for variables different from rainfall to define thresholds gives in principle more freedom. However, the basic idea is that the choice of the most suitable variable should be guided not only by pure statistical analysis (that is, how the threshold performs), but also, and mainly, by the identification of the hydrological processes that, for the kind of landslide and geo-morphological context, are expected to be responsible for "causing" the conditions predisposing to a landslide. The statistical analysis could be regarded as a tool to confirm if the process identification is correct or not.

*More details on these two points are given in the specific comments (comments to L253 and L 262-264). Finally, I recommend minor revisions for this manuscript.*

**Specific comments**

*L 20: "the conceptual idea is that precipitation information is a good proxy for bothmeteorological trigger and hydrological cause". It cannot be said that, in general, researchersderiving ID thresholds and their users have this conceptual idea in mind.This is a move of the authors which is not fully justified. So I think that this sentenceshould be rewritten, perhaps writing something on the fact that it is in general thoughtthat precipitation information can be linked by simple relationships to landslide occurrence,without explicitly taking into account hydrology.*

The reviewer is correct we of course cannot speak for all researchers/groups who derived ID thresholds that this was the conceptual idea. They also could have other justifications, like: it gives (statistically) good/useful results. We will rephrase to make clear it is our interpretation, and that the practical background of precipitation ID is that often only meteorological information is available when analyzing (non-) occurrence of shallow landslides, and that, at the same time, the conceptual interpretation of their success could be that precipitation is sometimes a good proxy for both meteorological trigger and hydrological cause.

*L 22: It is not fully clear what does "indistinct threshold" mean*

Agree. We will delete these words as indeed indistinct threshold is not defined. We will rephrase: "this approach suffers from many false positives …."

*L 36: "landslide is the most abundant hazard". Are the authors sure that "landsliding is the most abundant hazard"? Maybe say that it is "one of the most abundant naturalhazards", and add some references to literature (for instance: Sidle and Ochiai, 2013)Sidle, R. C. and Ochiai, H.: Landslides: Processes, Prediction, and Land Use, WaterResources Monograph, 2013.*

Agree, we will rephrase and add the reference for this:"one of the most abundant natural hazards"

*L 39 – 45: The three approaches listed by the authors are not all aimed to assess"landslide probability" in a strict sense (only number 3 is). In fact approach (1) leadsto an assessment of landslide "susceptibility", which is not exactly a probability, butan index of landslide proneness in a relative scale. Approach (2) does not provide ingeneral landslide probability, as most of the landslide triggering threshold schemes are"deterministic", and probability is in fact only in theory – but very seldom in practice– related to landslide triggering thresholds (Aleotti, 2004; Iiritano et al., 1998). Theauthors should clarify this point.*
*Aleotti, P.: A warning system for rainfall-induced shallow failures, Eng. Geol., 73, 247–265, 2004.*
*Iiritano, G., Versace, P., Sirangelo, B., 1998. Real time estimation of hazard for landslides triggered by rainfall. Environmental Geology 35 (2– 3), 175– 183.*

In a strict sense the reviewer is correct. We will replace 'probability' with 'possibility'. (and see comment L46 below).

*L 42: perhaps integrate literature on this, with other more recent papers (e.g. Peruccacci et al., 2017 and references therein)*
*Peruccacci, S., Brunetti, M. T., Gariano, S. L., Melillo, M., Rossi, M. and Guzzetti,F.: rainfall thresholds for possible landslide occurrence in Italy, Geomorphology, 290,39–57, doi:10.1016/j.geomorph.2017.03.031, 2017.*

Thanks for pointing to more recent literature.

*L 46: The term hazard may have a specific definition in the natural hazards field, related to the probability of the event to occur. So the authors should clarify that they refer to "hazard" in a broader sense. Perhaps in clarifying this they should cite a generally accepted definition of "landslide hazard". This comment is related to preceding one on L 39 - 45*

We are using the term (landslide) hazard in a broader sense, not in a probabilistic way: A (landslide) hazard is a natural phenomenon that might have a negative effect on people or the environment. We will clarify this in the text by adding a definition.

*L 63: the authors use both ID / PID when referring to precipitation intensity and durationthresholds. Only one way should be used*

Correct, we will use: (precipitation) ID

*L 63: "hazard" is perhaps not fully appropriate*

We will replace with: 'landslide hazard"

*L 70: add references to papers where a "probabilistic transition zone" is used*

We refer to Berti et al (2012) which is detailed on from L80 onwards

*L 88: It seems that authors are referring to works where antecedent precipitation isused (perhaps as a "measure of antecedent soil moisture content"). Here the authors should better clarify what they are referring to, and cite pertaining papers.*

We refer to measures indicating the wetness state of the soil anteceding a precipitation event triggering a landslide event. These are listed further on in the paper (L245 onwards). We will put some of those references here as well.

*L 88: It is unclear if antecedent precipitation should be seen in the authors' framework as an hydrological (cause) or meteorological (trigger) variable*

Hydrological cause: we detail on that from L245 onwards. We will add clarification here as well.

*L 90: again, here "hazard" is perhaps not fully appropriate*

Landslide hazard

*Figures 1 to 3: perhaps for a better comparison of the various curves it may be useful toplot in planes with the same axis range (e.g. x-axis of Fig. 1 goes from 0.1 to 100, whileFig. 2 from 0.1 to 1000). Also, it may be better that figures have the same appearance(e.g. no grid in the plot of Fig. 1; adjust font size in Fig. 3).*

Good suggestion, we will make the figures appear more homogeneous

*Figure 3: It is unclear how the dark grey area representing "landslide threshold" is derived from figure 2, as the area that it covers is narrower than that covered by thresholds in Fig. 2*

It is a generalised threshold summarizing figure 2. We on purpose narrowed the range for discussion reasons (like looking through eyelashes).

*L 171: It is unclear in which sense the ID threshold is "generalized"*

Broadly summarizing indication of the threshold.

*Figure 3: P is undefined (though its meaning can be easily understood from discussionin the text).*

As the reviewer suspects, P denotes precipitation, we will add this to the figure caption

*L 175: It is not clear why precipitation ID thresholds are "volumetric", as an infinitenumber of (I,D) or (H,D) pairs can be associated to a given event rainfall H.*

We mean to say: Every point on the threshold line depicts a volume, as H=I×D

*L 181: It is unclear why greater precipitation volumes should imply bigger landslides.Is this something reported in literature? I imagine that this is in general not true, asthe amount of rainfall gives little (or none) information on its spatial extension, and thusof that of the landslide. Also ID thresholds are derived using databases that usually report little information on landslide size, and to say that "the database consists for theoverwhelming majority of shallow landslides and debris flows" doesn't mean that the size of landslides is small.*

Sorry for the confusion, we are referring to deeper seated landslides: the amount of rainfall refers to unit surface area, so it cannot be related to landslide size, but it can be related to landslide depth. We will clarify.

*L 192 – 196: In this discussion the authors should mention that ID thresholds are sensitive to the way a rainfall event is defined, that is, mainly the maximum zero-precipitation interval within a rainfall event (See Vessia et al., 2014; Melillo et al; 2015). Cleary, the shorter this interval is, the shorter the length of rainfall*

*events will be. With long maximum dryness the events can be so long that different hydrological processes can takeplace. In this case rainfall events do not represent "the last push" but a mixture between "causes" and "triggers".*

*Vessia, G., Parise, M., Brunetti, M. T., Peruccacci, S., Rossi, M., Vennari, C. and Guzzetti, F.: Automated reconstruction of rainfall events responsible for shallow landslides, Nat. Hazards Earth Syst. Sci., 14(9), 2399–2408, doi:10.5194/nhess-14-2399-2014, 2014.*

*Melillo, M., Brunetti, M. T., Peruccacci, S., Gariano, S. L. and Guzzetti, F.: An algorithmfor the objective reconstruction of rainfall events responsible for landslides, Landslides, 12(2), 311–320, doi:10.1007/s10346-014-0471-3, 2015.*

This is our point. We do discuss this point in L161-L165. However, we agree with the reviewer that this point should be highlighted here as well. Thanks.

*L 253: The authors should discuss how to separate between the time scales of "causes" and those of the "triggers". In other words, how to switch, in practice, from the "cause" hydrological analysis (storage), to the "triggers" meteorological analysis (rainfall)? In other words, how does the framework the authors propose contribute in removing the subjectivity of identifying the rainfall that represents the "trigger"/"lastpush" (see comment on L 192 – 196)?*

See reply with general comments

*L 253: Another point is: hydrology may be in general important also during the "triggering" process, while in the authors' framework it is not explicitly taken into account. Arethe authors implicitly saying that "hydrology of the last push" can be taken into account without a significant processing of rainfall data?*

The point we want to make here is that we propose that an effort is made to separate longer time scale causes with shorter time scale triggers. Of course we do not imply other mechanisms cannot take place, but processes evolving over shorter scales are more directly related to the characteristics of rainfall event (i.e. the intensity), while rainfall effects on long-term processes are smoothed.

*L 262-264: "However there are several possible choices of hydrological variables to be plotted along the cause-axis, such as soil water content, catchment storage, representative regional groundwater level and similar". This implicitly reveals that a high degree of subjectivity follows from the framework that the authors propose. Researchers do generally agree that subjectivity of the ID threshold assessment is significant, in spite of his simplicity. For instance, one source of subjectivity in ID thresholds is related to the choice of the maximum zero-precipitation interval to define rainfall events (seecomment on L 192-196). This is known to impair comparisons between thresholds, which thus makes it difficult to search for general landslide triggering thresholds. The framework that the authors propose seems to possibly bring a higher heterogeneity of the analyses, and thus maybe can in practice represent a step backwards for finding unifying concepts. By introducing hydrological analysis, researchers may have more freedom in choosing models and parameters for estimating the "cause" variable (antecedent soil water content). This may represent a possible way to manipulate theresults so that the performances of the resulting hydro-meteorological thresholds appear to be higher than they actually are. Thus, the authors should discuss how one can prevent this, perhaps by highlighting the importance of always performing validation analyses, i.e. to test developed thresholds against a sub-dataset which is not used in calibration.*

The point we make is that (hopefully) more process knowledge will be in the graphs and less statistics. Indeed there will be a higher degree of freedom from a statistical point of view, but we argue also higher causality. The reviewer is right that this subjectivity can be a problem, but we take the stand that searching for a hydrological cause will improve the physics behind the thresholds whereas now, we rely only on statistics.

*L 319: "ID thresholds neglect the role of the hydrological processes" is a strong statement. Indeed it may be written that hydrological processes are too simplistically represented by ID thresholds. In other words, precipitation is the main cause of landslides, but the main problem is: how to process precipitation information to obtain thresholds that perform well in forecasting landslides? And, of course, ID thresholds certainly do not represent the best way to processes rainfall data.*

We do not agree with the reviewer that this statement is too strong. There is no attempt to put hydrology into the threshold and only in case of relatively linear relationships between hydrology and precipitation this type of thresholds would hold. But hydrology (pore pressure built-up in the subsurface), is not linearly related to precipitation. So, yes, we can look at ID thresholds in terms of statistics, but we argue one could also look it from the point of "being right for the right reason".

*L 332: I agree that one downside of spatially-distributed physically based models is that they require a "well calibration". However to estimate catchment storage (as in Ciavolella et al., 2016), requires a well calibrated model too. The authors should discussbetter this point.*

Point well taken. Indeed, in practice also some of calibration will remain required, although to a lesser extend then calibration of a fully distributed model. We will add this in the discussion.

*L 233: A sketch explaining the approach the authors propose can be useful for readers.*
Good suggestion, we will try.

**Technical corrections**

We are thankful for these technical corrections. We will correct them in the revised version
*L 42: Caine instead of Cain*
*L 45: maybe something is missing as citations finish with a ";"*
*L 71: "separation" instead of "separator"*
*L 78: remove "," after "conditions"*
*L 142: perhaps replace "for regions or areas not pertaining to this area" with "otherregions or areas"*
*L 147: "threshold" instead of "thresholds"*
*L 197: perhaps "phenomena" instead of "hazards"*
*L 198: "related" instead of "relate"*
*L 223: "thresholds" instead of "threshold"*
*L 255: perhaps "field" instead of "terrain"*
*L 283: "specific" instead of "particular"*
*L 318: "limitations" instead of "limitation"*
*L 326: "interpretations" instead of "interpretation"*
*L 331: "physically based" instead of "physically-based"*

---

## Author Comment (AC2) · 23 Oct 2017

**Nat. Hazards Earth Syst. Sci. Discuss., https://doi.org/10.5194/nhess-2017-241**
Invited perspectives. A hydrological look to precipitation intensity duration thresholds for landslide initiation: proposing hydro-meteorological thresholds

**Reply to Referee # 2; Roy Sidle**

We thank Roy Sidle for the detailed and constructive review. Also his insights in original literature is very valuable. Thanks so much. Below we address and reply to the comments and questions. In Italic typesetting the original review is given, and in roman typesetting our replies.

Thom Bogaard and Roberto Greco

*This paper offers a refreshing and needed critique of the ID relationships commonly used in regional (and global) landslide predictions. Furthermore, it proposes an improved approach that includes metrics of predisposing 'causes' and 'triggers' of shallow landslides. As such, it should stimulate new research in this arena that will benefit landslide and hillslope debris flow prediction. It will be a valuable contribution to NHESS with some moderate revision.*

*I noted several times in my review that follows that ID relations (at least some previously published ones) have erroneously reported data as 'individual storms', which were obviously not individual events (i.e., very, very long durations). Additionally, on the opposite side of the ID 'x-axis' there are instances of very, very short storms of high intensity triggering landslides – these appear to be bursts of intensity on saturated soils as noted by the authors or they could in fact represent a totally different process, like channel bed mobilization causing a debris flow. My recollection of reading through some older reports in which data were used to develop ID thresholds is that in some cases the described mass failure was more of a within channel debris flow. Off the top of my head, I am thinking of some of Rapp's early papers that were included in Caine's threshold. In any event, these anomalies should be considered or mentioned herein.*

The reviewer clearly indicates the importance of rainfall variation over a rain period. This indeed is extremely important as also discussed in our paper (L163-L166; L327). However, we agree, we should discuss this part more. We will add a new paragraph around L163 in which we detail on the effect of a) the uncertainty arising from precipitation measurements using gauges and b) the effect of using average intensities in the ID thresholds and that when moving from short to very long events it is not possible to have identical definitions of what an event is.

*I have noted a number of editorial suggestions directly on the manuscript which I will attach for the authors.*

Thanks, we are thankful for these and will benefit from them.

*More scientific technical comments are noted as follows:*

*Title: I would say "Hydrological perspectives on precipitation intensity – duration thresholds. . ..."*

The title starting with "Invited perspectives:" is a format of NHESS. We will discuss the title based on your suggestion (and of Ben Mirus) with the executive editors of NHESS.

*Lines 28-29: reword – "discuss" based on "associated discussion"*

OK

*Lines 86-88: yes, we tried this in our 1985 Hillslope Stability and Land Use book using antecedent rainfall information, but the problem was the lack of documentation of such antecedent rainfall data in earlier studies. Overall, we felt that it did improve the ID thresholds (at least conceptually).*

Thanks for sharing, we will make appropriate reference to it.

*Line 109: The term 'stormwater management' implies to me more of an urban planning context; that may be my bias, but you may want to add 'flood prediction' (or something like this) as well.*

We indeed use the term for urban stormwater management in this context. We will add the flood prediction.

*Line 125-126: I think that this is a key difference between practical applications of IDF and ID curves; that is most (or at least many) shallow landslides respond to sort-term intensity bursts which are not articulated in typical IDF's. You may want to mention this.*

Agree, indeed the rainfall durations considered for the derivation of IDF curves have nothing to do with the beginning or with the end of a rainfall event: they are time intervals of given duration during which some rainfall fell. And then, extreme observed values are considered (e.g. annual maxima). Shallow landslide tend to respond indeed to short term high intensity of rainfall which are often not visible anymore when averaging over larger time steps. We will include this aspect in our paper (see also below)

*Line 144: Try not to start sentences with "Figure x shows. . ."; this can be seen in the Figure and caption. Just directly say what you wish to say about the data in the figure and cite the figure in parenthesis at the end of the sentence.*

We will reformulate this.

*Lines 155-157: rework this sentence – understandable, but a bit confusing. Maybe just put 'mostly debris flow and some shallow landslides' in parenthesis. Furthermore, I think there are some issues with such very short 'landslide producing storms' reported in the literature that are captured in these cited thresholds. As you note, they are probably mostly debris flows, and upon inspection of some earlier papers that reported such short-term events, it seemed that the authors were referring to possibly a different process – e.g., debris flows caused by channel bed mobilisation. I looking into this matter in our 2006 landslide book, we actually threw out some of these short-term rain events when constructing new ID curves because we were convinced that they represented different triggering processes.*

We rephrase as follows: "For landslides triggered by short precipitation events (D ≤ 1 hr), the slopes of the IDF and ID curves substantially coincide (Figure 3). ".

Furthermore we will add it after line 161: "This is counter-intuitive, as during long-lasting wet periods landslides are usually more frequent, while many debris-flows triggered by very short and intense storm originate from channel bed mobilization rather than being (new) mass movements"

*Lines 158-166: I agree that this is problematic, and I feel (as you state) that ignoring short-term peaks of rainfall in an otherwise long-duration, lower intensity event is the main reason for this problem. Based on my work and that of others, I always say that one common scenario for shallow rapid landslide initiation is a long storm of low to moderate intensity, with a peak intensity occurring near the end of the event. Another issue here, I agree that the longer return periods for landslides triggered by long-duration, low intensity storms is counterintuitive; however, when we looked into the actual data for some of these so-called long duration events that triggered landslides (in reviewing references for the 2006 AGU landslide book), it became apparent that some of the data included in these ID relationships were not strictly 'individual events', rather these were based on a longer period of rainfall leading up to the landslide. – thus, a direct comparison with some of these so-called long-duration landslide triggering 'events' with IDF curves for actual individual events may be a bit problematic. You probably should mention this potential discrepancy. My point is probably only relevant for the very long 'events', but it may be worth mentioning.*

(See also reply after general comment at top and after Line 125-126) The importance of rainfall intensity variation should be elaborated on, and as stated before, we will add a paragraph discussing this in more detail.

*Lines 185-188: This sentence is a bit confusing; it seems that you are referring to reported data when you saw the 'vast majority of empirical thresholds fall between . . .". Are you saying that for other studies most of the landslide reported would fall between thresholds of 10 to 100 mm? If so, you need to cite some references. But I am not sure that is what you are trying to say here. Anyway, please clarify. (and you overuse the expression 'vast majority' – just say most or the majority).*

We reformulate: "Many of the reported empirical precipitation thresholds has between 10 and 100 mm of accumulated precipitation. However, also <10 mm and >1000 mm volumes needed for landslide initiation have been reported."

*Lines 194-196 See my previous comment about data for very long duration 'events' that are likely not individual events.*

*Lines 209-210: In addition to my comment in the text, also see my previous comment about data for very long duration 'events' that are likely not individual events.*

Agreed. See response above

*Lines 227-231: Very complex sentence and a bit awkward. Can you rewrite this or try to break it up a bit?*

We reformulate: "Hence, it was possible to define non-dimensional variables comparing the meteorological triggers with the infiltration and storage capacity of the soil cover. This non-dimensional hydro-meteorological threshold performed slightly better than the precipitation ID threshold in separating events resulting in factors of safety smaller and greater than 1.3."

*Lines 240-241: I don't mean to be beating a 'dead horse' again, but such long events are obviously not 'events'; they were probably included in databases because this was the only precipitation record reported.*

Agreed. See response above

*Line 257: Why do you say 'was preferred'? by who?*

We intended to express that in several studies another approach was followed, that of distributed physically-based modelling. We will change in: "This is a largely unexplored terrain, although we recognize that data availability can be cumbersome. " The discussion on the option of using physically-based distributed models is done elsewhere in the paper (In introduction and in conclusion L330).

*Lines 258-260: Reword the first sentence to note that the trigger axis refers to the rainfall characteristics (intensity) responsible for initiating the landslide. When you say "depends on the local situation" – I think you mean both available data and the rainfall characteristics that are responsible for landslide initiation in that area.*

Reformulated as: "Concerning the 'trigger'-axis, there is little debate; it is the rainfall intensity responsible for the short-term last push initiating a landslide."

*Line 276: What do you mean by 'discharge intensity'? This is a rather unconventional term.*

Reichenbach et al (1998) used: event intensity (in $m^3 sec^{-1} km^{-2}$). We will explain and use "specific discharge".

*Line 282: What is low/high storage?*

We mean to say how much water is stored in a catchment compared to its maximum storage capacity. We clarify.

*Lines 286-287 (and the sentences that follow): I think this phenomena occurs for deep seated landslides like earthflows of slump-earthflows – maybe better to state this to avoid confusion, because you are mostly focussing on shallow landslides. There is some older work in Japan that has clearly showed such relationships with earthflow reactivation and a threshold groundwater depth. I believe mark Reid also published a paper on this from earlier work in Hawaii.*

Thanks, indeed this mainly holds for deeper seated landslides. Thanks for the reference. We will rephrase: "In some cases, mainly deeper seated landslides, ….."

*Line 322: Again, not all data in these ID relationships were for individual events.*

Agree

*Line 323: you mean even when they are developed for small areas?*

Yes. Thanks

*Line 338: These will be particularly valuable in developing countries.*

True!

*I really like the message in the last paragraph of the Conclusions! Well articulated.*

Thank you

---

## Author Comment (AC3) · 23 Oct 2017

**Nat. Hazards Earth Syst. Sci. Discuss., https://doi.org/10.5194/nhess-2017-241**
Invited perspectives. A hydrological look to precipitation intensity duration thresholds for landslide initiation:
proposing hydro-meteorological thresholds

**Reply to Referee # 3 Francesco Marra**

We thank Francesco Marra for the insightful comment. Below we address and reply to the comments and questions. In Italic typesetting the original review is given, and in roman typesetting our replies.

Thom Bogaard and Roberto Greco

*The authors analyze the concept of precipitation intensity-duration (ID) thresholds for shallow landslides and debris flows from a hydro-meteorological perspective to propose a new approach to the problem. The contribution is largely welcome since it suggests new approaches and perspectives to overcome important, but sometimes neglected, limitations of the commonly used methods.*

*Within the interesting analysis of the relationship between ID thresholds and IDF curves, the discussion so far neglects the impact of rainfall estimation uncertainty and its possible dependence on rainfall duration and return period (e.g., Krajewski et al., 2003, www.dx.doi.org/10.1623/hysj.48.2.151.44694; Ciach and Krajewski, 2006, www.dx.doi.org/10.1016/j.advwatres.2005.11.003). Recently, rainfall estimation uncertainty caused by the use of rain gauge measurements (still the most common source of rainfall estimates in this field) was shown to significantly affect the derived ID thresholds causing systematic bias (Nikolopoulos et al., 2014, www.dx.doi.org/10.1016/j.geomorph.2014.06.015). This systematic bias is caused by (i) systematic rainfall patterns observed around the triggering locations and (ii) the use of log-transformations within the derivation of the ID (Marra et al., 2016, www.dx.doi.org/10.1016/j.jhydrol.2015.10.010). At least for durations 2 days, these rainfall patterns were observed to be related to the return period of the triggering rainfall (Destro et al., 2017, www.dx.doi.org/10.1016/j.geomorph.2016.11.019). Consequently, the 'slope' of the ID threshold is affected by rain gauge sampling, and this potentially undermines the comparison between the slope of ID thresholds and IDF curves reported in the manuscript. In particular, I think this aspect should be discussed when the authors say [line 157 and following]: "On the other hand, for longer precipitation durations, ID thresholds have smaller slopes than IDF curves.This means that landslide initiation on the right side of the graph (lower precipitation intensity with longer duration) would occur with rapidly increasing return periods of precipitation events". In fact, the observed pattern could be caused/emphasized by the sampling issues discussed by Marra et al., (2016) and Destro et al. (2017).*

*To conclude, IDF curves are expected to vary within the examined region (generally a regional, if not global, scale); it is thus unclear to me how the idealized IDF curves in Fig. 3 have been drawn. The 'slope' of the curves (i.e. the dependence of I with D) at the regional scale may change so that the curves should be better represented as a shaded area – such as done for the ID thresholds.*

*This was a technical comment on an introductory aspect of the study; this being said, I repeat my compliments to the authors for the manuscript and the new proposed perspective.*

*With kind regards, Francesco Marra*

Dear Francesco Marra

Thank you very much for you excellent comment. We strongly focus on the hydrology whereas you bring up the point of the effect rain patterns and rain measurements (rain gauge based precipitation observation, spatial organisation of rain and associated uncertainty) that so far we on purpose have left out of the article. We tried to keep the article brief as is requested for an "invited commentary". Our focus is that only using precipitation data for landslide hazard assessment is neglecting the importance of hydrological processes in landslide initiation. Hereto, we make a quite extensive 'problem' analysis showing the currently used ID thresholds have limited physical meaning, at the best they are statistically interesting to use. However, you are of course fully right that precipitation estimates and uncertainty can also be responsible for (part of) the observed low slopes of the ID thresholds. So, in hindsight we agree this is a too important aspect not to address, so we will add a short discussion on the effect of precipitation measurement uncertainty as depicted in your comment.

Your second point is about the (slope of the) IDF curve that we published in Figure 3. Yes, this is a somewhat arbitrary but representative, set of IDF curves (that is, the chosen slope is not far from those of the examples of IDF curves of Fig. 1, which refer to very different locations around the world). Like we write in the caption "Schematic" and in our opinion useful for the argumentation. It is however a good suggestion to show, also graphically, in Figure 3, that the IDF are schematic, so we will adapt the figure and explain it in the text as well.

---

## Author Comment (AC4) · 23 Oct 2017

**Nat. Hazards Earth Syst. Sci. Discuss., https://doi.org/10.5194/nhess-2017-241**
Invited perspectives. A hydrological look to precipitation intensity duration thresholds for landslide initiation: proposing hydro-meteorological thresholds

**Reply to Referee # 4;** *B. Mirus,* bbmirus@usgs.gov

We thank Ben Mirus for this pro-active, stimulating and detailed review of our discussion paper. Below we address and reply to the comments and questions. In Italic typesetting the original review is given, and in roman typesetting our replies.

Thom Bogaard and Roberto Greco

*The authors present a much needed discussion about some systematic problems with precipitation intensity-duration (ID) threshold approaches for predicting shallow landslides. In particular, they point out an unfortunate lack of reasonable constraints on the max/min duration of rainfall events, and also discuss how these unbounded events can affect the average intensity and predictive capabilities. Both issues are largely ignored in many studies focused on developing and testing ID thresholds, so it's a worthy discussion about some crucial sources of error. The authors also highlight the potential importance of hydrological information in addition to precipitation characteristics, which have not been systematically incorporated into landslide early warning criteria. In my opinion the most innovative contribution presented in the manuscript is the comparison of the rainfall intensity recurrence intervals and ID thresholds for landslide initiation from the literature, along with the contour lines of cumulative storm totals (Fig. 3). This is a new and intuitive way to broadly illustrate their point about some problems with the ID threshold concept.*

We agree that Figure 3 is key in our argumentation of the issues linked to precipitation ID thresholds.

*However, my primary concerns with the manuscript are twofold:*
  *(1)   Limited concrete guidance is provided on how to apply the proposed "cause-trigger" framework, so the potential novelty of the approach seems somewhat overstated. The general concept that both the predisposing factors (e.g. antecedent wetness) and a rainfall triggering event are needed to explain shallow landslide initiation is already generally accepted and has in fact been implemented in a number of landslide initiation thresholds. For example, two different rainfall thresholds developed for the Seattle area explicitly account for antecedent factors: (a) the recent-antecedent cumulative precipitation threshold compares the 3-day triggering rainfall to the 15-day antecedent rainfall (Chleborad et al, 2008), and (b) the Antecedent Water Index is used with an exponential ID threshold for events between 10min and 10days in duration, though storms are generally less than 24hours (Godt et al., 2006). The authors also cite several other papers (including some of their own published work) that in various ways incorporate antecedent wetness as a measure for the predisposing factors prior to the triggering rainfall event using soil water balance modeling or catchment storage. As such it is not clear how the "cause-trigger" approach is truly novel, but rather seems to be a new term for a topic in need of further exploration. (As an aside, the term "predisposing factors" seems to be a more appropriate term: without context "cause" could be misleading since both predisposing factors and a triggering event are needed to cause a landslide.)*

The objectives of this invited perspective are to: (a) critically analyse the precipitation ID thresholds for shallow landslides and debris flows from a hydro-meteorological point of view; and (b) propose a conceptual framework for lumped hydro-meteorological hazard assessment based on the concepts of trigger and cause.
We think we do not claim to be 'truly novel' in our paper. Obviously, as also discussed in the paper at length, we write this perspective-paper standing on shoulders of giants; the many colleagues who have shed light on this topic before us. In our article, we brought it into an overarching conceptual framework under which new research could (should) be done (L340-L350), also because many of the existing examples - like the two you point out - look for the causal relationship between antecedent precipitation and landslide occurrence merely by maximizing some measure of correlation, while our proposed framework invites to look at the most relevant hydrological process for the considered context. Only in the abstract (L27) we did write "novel trigger cause concept". Here we will reformulate in "we propose a trigger-cause conceptual framework"

The framing into 'trigger-cause' and then especially the "cause" is indeed debatable We defined the word "(hydrological) cause" as the predisposing (hydrological) condition of an area under study (L260, L320). We will define our use of the word "cause" earlier in the paper (section 3).

> *(2) The critical issue of data availability is understated.The topic of data availability is largely avoided until the very end of the conclusions, at which point it comes across as an afterthought instead of the main reason the precipitation ID threshold has been employed successfully for decades. Without continuous records of appropriate data during historic landsliding events it is challenging (if not impossible) to develop and test alternatives to the precipitation ID threshold. The reality is that rainfall data is widely available and has been for some time, which has facilitated useful, albeit somewhat flawed tools for assessing landslide potential for a number oflandslide-prone areas. Secondly, rainfall can be predicted in advance with considerable accuracy, so despite some errors in ID thresholds, the trade-off between appropriate lead-time using weather forecasts and threshold accuracy must at least be considered when arguing for alternative threshold approaches. Without a more balanced discussion of data availability it's not entirely clear whether the authors are arguing for better analysis of rainfall data that distinguished between the "causing" rainfall and the "triggering" rainfall or if the authors suggest that rainfall is not an appropriate data source for the "cause" variable and the ID threshold concept has been employed incorrectly for very long and very short duration storms. Although the Invited Perspective highlights both these problems with the ID threshold approach, it remains unclear how the"cause-trigger" approach can be used to solve these problems within the context oflimited data availability.*

Indeed, we decided to first set-up the problem, then the concept and lastly, the "afterthoughts". However, we do mention the issue of data availability several times (e.g. L255, relatively in the beginning of section 3). We do agree that the data availability is the issue (at the moment) and we agree to stress that in our paper also in writing why the traditional ID thresholds are often preferred/used. May we add that once the driving hydrological process is identified, modelling can supplement the lack of data, while this is impossible with rainfall alone?
The fact that rainfall data is widely available is, in our opinion, maybe one of the reasons for misuse of ID thresholds. I our opinion "somewhat flawed" seems an understatement. Could it not be that relying on rainfall records prevented us from digging deeper? Secondly, many papers address the issue of how representative this rainfall information is. By 'uncritically' linking landslide occurrence to the nearest precipitation record, we seem to prefer practical/statistical correlation over causal relation. In this perspective, this is one of the two aims.

> *After addressing these two issues regarding the novelty of the proposed approach and the availability of data for landslide initiation thresholds, the authors' hydrologic perspectives on the precipitation ID approach for landslide prediction will be a valuable contribution and will surely be of considerable interest to readers of NHESS. I suggest a number of general and specific revisions prior to publication, outlined in the sections below. Thank you for your consideration.*
> *Kind regards, Ben Mirus*

> *General Revisions:*
> *(a) Explain how the details of how the "cause-trigger" concept can be distinguished from prior contributions that consider antecedent conditions, or qualify the novelty ofthe proposed approach within the context of such prior work.*
> *(b) Provide more concrete guidance on how the "cause-trigger" framework could beapplied for future studies. In particular, how should researchers constrain the durationof storms to distinguish between "cause/predisposing factors" and "trigger"?*
> *(c) Include a more balanced discussion of what data could reasonably be obtained toinform the "cause" axis for any landslide early-warning threshold relative to the widelyavailable (and forecastable) input of rainfall.*

See above

> *Specific Edits:*
> *L1: I agree with the revision to the title suggested by Roy Sidle and would further suggest removing the second phrase since there no specific hydro-meteorological thresholds are proposed. Thus the suggested title is shorter and more precise: "Hydrological Perspectives on the Precipitation Intensity Duration Thresholds for Shallow Landslide Initiation"*

The format of the title is set by NHESS. We will discuss this with the editor in charge. We agree: Hydrological perspectives etc is to-the-point.

*L13: Provide some citation or definitive evidence for the strong, yet disputable statements like "vast majority" and "never" . . . otherwise such pronouncements should be avoided in scientific writing. Furthermore, as is later argued in the manuscript, precipitation does not actually initiate the landslide. Suggest revising to: "Many shallow landslides and debris flows are rainfall induced."*
The intention of an invited perspective is also to raise discussion and have sharp edges. We therefore allowed ourselves a writing style which is "less scientific" as generally seen in scientific articles. But as all reviewers bring this point on, we clearly 'overdone' it and we will discard such strong pronouncements.

*L22: What does "indistinct" mean? Thresholds are by nature distinct. On the other hand, the errors resulting from application of distinct thresholds over broad areas reflects the heterogeneity of natural systems. In theory, each hillslope/hollow has a unique threshold that must be averaged over some area and some time to create auseful tool for landslide early warning.*
Agree. We will delete these words as indeed indistinct threshold is not defined. We will rephrase: "this approach suffers from many false positives …."

*L27: Again, calling this a novel conceptual framework is an overstatement. See general comments.*
See reply above. Will change in: A conceptual framework

*L36: References to support this claim? I was not aware landsliding is the MOST abundant hazard. At the very least it should be qualified as a natural hazard, sincemany health or other hazards could be considered more abundant and/or detriment also socio-economics.*

Agree, we will rephrase and add reference for this: "one of the most abundant natural hazards"

*L40-43: Unclear from the description provided here how 1) and 3) are different when applied to assessing landslide probability. Perhaps some example citations later in the paragraph could help distinguish between the two.*
See also Reviewer #1: We will replace 'probability' with 'possibility'

*L59: I recommend also citing Anagnostopoulos et al. 2015 when discussing model complexity.*
Thanks for the suggestion.

*L74-75: Perhaps include some more recent citations that are less than 10 years old?*
You have a point here, we will add more recent references.

*L78-79: Yes. Also there is considerable error introduced by the heterogeneity that must be "averaged out" for a PID threshold to be developed over an area of interest.*
Yes

*L86-88: Are you proposing something that is better than soil moisture? It seems that soil moisture would be better than the other variables suggested (albeit harder to measure),so it is seems counterintuitive to state that these studies are "limited" to measuresof antecedent moisture content.*
Correct, "limited" should be removed: "however they were mainly including measures of antecedent soil moisture content, which may not represent the most suitable variable for any kind of landslide"

*L88: Never say never. In general it is unwise use this word in scientific writing unless it can be rigorously confirmed, which is almost "never" possible. Suggest revising to"not" or "have not been the subject of"*
As replied before we have been somewhat over-enthusiastic in our writing style. We agree to reduce the use of the strong pronouncements.

*L106: Here and elsewhere the abbreviation switches from PID to ID. . . either is fine,but use only one consistently throughout.*
Right, we will use ID

*L149: What is an "absolute" value of a threshold? Do you mean, for example the xandy-intercept values? Revise for clarity.*
*L152: Again, what is the "absolute" precipitation ID?*
We mean unscaled measured precipitation values

*L153: At some point in this part of the discussion you should mention the novel use of duration-frequency curves by Fusco et al., 2017. They use this concept to examine temporal and spatial patterns of pressure head states that predispose slopes to recharge and/or landslide initiation. I think it is a nice example of how the "cause" concept you propose could be implemented practically.*

Good suggestion. Thanks for pointing us to it. This is indeed a nice example.

*L167: Yes. This also leads to questions about how storm durations are defined, particularly since longer storms are more likely to include actual breaks in precipitation where drainage and ET can be more effective in reducing landslide initiation potential.*

*L181-184: Revise these sentences for greater clarity. It seems like the main point is that if larger cumulative precipitations are needed to initiate landslides at lower intensities we would expect that to be reflected by the larger landslides in the inventory, but the inventory you reference is mostly small landslides, so an alternate interpretation is that the slope drains while it's raining? At least the last sentence is incomplete to communicate the message more clearly.*
Sorry for the confusion. The amount of rainfall refers to unit surface area, so it cannot be related to landslide size, but it can be related to landslide depth. Indeed, we are referring to deeper seated landslides. We reformulate.

*L186: Technically this (between <10 and >1000) is not a range, it is unbounded. Do you mean between >10 and <1000? Revise for accuracy.*
We reformulate: "Many of the reported empirical precipitation thresholds has between 10 and 100 mm of accumulated precipitation. However, also <10 mm and >1000 mm volumes needed for landslide initiation have been reported."

*L187: Again, such strong statements like "vast majority" should be supported by a number of independent citations or other evidence. Otherwise avoid this term.*
Agree

*L200: Not sure this is the most appropriate phrasing. The real utility of ID thresholds is that they are not at all cumbersome to use, but rather involve a very simple and easy interpretation: does the rainfall intensity and duration plot above or below the threshold line? Maybe more important point is that ID thresholds applied locally need a "calibrated range" for storm duration whereas regionally and globally they are misleading since there is too much spatial variability in rainfall and hillslope hydrologic responses for accurate predictions.*
Good point, the ID thresholds are maybe too easy to use: We change the word "use" into "interpretation". This makes the interpretation of ID thresholds cumbersome.

*L206: Napolitano et al., 2015 is another good reference to include here as they also used seasonal variations in antecedent soil wetness to identify different thresholds for winter vs. summer.*
Thanks for the suggestion. We would also add some remark about the fact that, when this kind of thresholds accounting for previous precipitation (seasonal variation) have been proposed, then the considered antecedent duration is usually the mere result of a correlation analysis. By exploiting process knowledge, instead, you directly can look at the most appropriate physical cause-effect relationship (and thus variable).

*L218-219: Indeed, this is the concept underlying the recent-antecedent cumulative rainfall threshold of Chleborad et al., 2008, except they use prescribed durations, which have since been statistically tested with receiver operator characteristics (Scheevel et al., 2017)*
Thanks for this insight. We add this.

*L229-231: The wording of this sentence is confusing. What is the significance of this separation between near-failure events at FS < 1.3? Without reading the papers listed before it's not really clear why this is relevant.*
We reformulate: "Hence, it was possible to define non-dimensional variables comparing the meteorological triggers with the infiltration and storage capacity of the soil cover. This non-dimensional hydro-meteorological threshold performed slightly better than the precipitation ID threshold in separating events resulting in factors of safety smaller and greater than 1.3. The choice of referring to a factor of safety larger than 1.0 was dictated by the actually observed soil conditions during the monitoring period"

*L247: Not exactly, for such studies soil water content is not usually measured directly. Suggest revising to "proxy" instead of "measure".*
Good suggestion. We will use "proxy"

*L261: Again, I much prefer the term "predisposing condition" (or factors) over "cause" since both the trigger and predisposing factors essentially conspire to "cause" the increased pore pressures and reduced strength that initiates a landslide.*

Although we see your point we prefer the word "cause" (defined as "predisposing (hydrological) condition"), because it is a way to highlight that we are looking for the identification of the causal hydrological process.

*L263: Although perhaps beyond the scope of this paper, the "cause-triggers" approach ought to be more universally applicable to landslides, including those triggered by earthquakes or erosion. For example, an earthquake may trigger more landslides in wet vs. dry soils. Similarly, erosion at the toe of a slope may allow it to become more predisposed to failure during a rainfall event. Perhaps worth considering as you explore and develop this framework in the future.*
Yes, this is indeed an outlook we have. And yes, your examples are excellent. I think this is a very good suggestion for future work.

*L264: What kind of "storage" is this? Do you mean how much water is stored in the catchment (e.g. an effective saturation)? Or do you mean how much water the catchment can store (i.e. storage capacity)?*
We mean to say how much water is stored in a catchment compared to it maximum storage capacity. We clarify

*L286: How is a catchment itself more or less permeable? Bedrock permeability is clearly a measure you could consider, but very permeable bedrock would result in all groundwater recharge and no runoff (or landslides?). Perhaps more relevant would be the thickness and hydraulic conductivity of the soil, which ultimately are reflected byhow quickly the catchment drains.*
Indeed not well formulated, it indeed links to soil hydraulic characteristics. Chitu et al found that two catchment with low infiltration capacity and thus larger direct runoff fraction links to landslides whereas in the other catchment the underlying hydrological process was infiltration and pore pressure build-up. We will reformulate.

*L290-291: There are a lot of things mixed up in here, which makes it difficult to relate to the primary topics of the article. First, this is not a shallow landslide, so perhaps this is a bit of a tangential argument for this paper, but it seems that the main point is mobility for a large, slow moving landslide can be related to groundwater levels. That's fine.However, it's not clear groundwater levels would be a good proxy for conditions favouring shallow landslides, particularly since deeper groundwater levels might not respond until after shallow soils on hillslopes have drained and are no longer susceptible to failure.*
Yes, you are completely right here, this example is not a shallow landslide, but indeed a deeper-seated, reactivating, coastal bluff type of landslide. This will be explicitly mentioned, and yes, regional GW does not need to be a good proxy for shallow landslides but we also did not claim that either. But we agree this was not clear. Thanks for pointing to this.

*L310-314: OK. This makes sense, but can you provide more concrete guidance or framework for evaluating the appropriate "cause" variable? Also, can you provide some balanced perspective of how readily available those types of data may be relative to rainfall? An example Figure 4 might be helpful.*
In this perspective we aim to give direction to future research by providing a problem analysis and an overarching framework. Cautiously, we also give some examples, and we later on discuss that the approach will (currently) suffer from data availability. In my own experience, we ran into this problem as well when searching for hydrological information to use. However with more and more data coming available, we argue the community could try to make the step from "statistical/practical' threshold to "causal relationships".

*L322-323: This is a somewhat subjective (i.e. value) judgment, which is tangential to the discussion presented here. The perceived or tangible value of predicting even 1/100 landslide events correctly at the expense of many false alarms is an entirely different question. Probably "predictive accuracy" would be more appropriate.*
Thanks for the suggestion. We will change into "predictive accuracy"

*L340-349: I completely agree with these statements and don't wish to argue with the sentiment, but at the same time the conclusions are rather wordy and not particularly satisfying or informative. Another (shorter) way of saying this is that hydrologic information could improve individual thresholds for shallow landslide initiation, but the type of hydrologic information that is most appropriate will vary based on location and data availability. So then how do we go about addressing this issue?*
Indeed the last paragraph is somewhat wordy. One reviewer likes it a lot, the other one a bit less. We will try to use shorter sentences in the last paragraph to increase readability.

*L350: The last-minute mention of remote sensing comes across as a bit of an afterthought and it is not clear how this very broad suite of information products can be used to constrain hillslope water balance. Why not soil moisture monitoring?*

Indeed is the explicit mentioning of RS unnecessary and an "afterthought". We will replace by mentioning the increasing hydrological data that become available (without specifying).

References Cited:
Anagnostopoulos, G.G., S. Fatichi, P. Burlando, 2015, An advanced process-based distributed model for the investigation of rainfall-induced landslides: The effect of process representation and boundary conditions, Water Resources Research,doi:10.1002/2015WR016909
Chleborad, A.F., R.L. Baum, J.W. Godt, P.S. Powers, 2008, A prototype for forecasting landslides in the Seattle, Washington, Area, Reviews in Engineering Geology, doi:10.1130/2008.4020(06).
Fusco, F., V. Allocca, and P. De Vita, 2017, Hydro-geomorphological modelling of ashfall pyroclastic soils for debris flow initiation and groundwater recharge in Campania(southern Italy), Catena, doi:10.1016/j.catena.2017.07.010.
Godt, J.W., R. L. Baum, A.F. Chleborad, 2006, Rainfall characteristics for shallow landsliding in Seattle, Washington, USA, Earth Surface Processes and Landforms, doi:10.1002/esp.1237.
Napolitano, E., F. Fusco, R. L. Baum, J.W. Godt, P. De Vita. 2015, Effect of antecedent hydrological conditions on rainfall triggering of debris flows in ash-fall pyroclastic mantled slopes of Campania (southern Italy), Landslides, doi:10.1007/s10346-015-0647-5)
Scheevel, C.R., R.L. Baum, B.B. Mirus, J.B. Smith, 2017, Precipitation thresholds for landslide occurrence near Seattle, Mukilteo, and Everett, Washington: U.S. GeologicalSurvey Open-File Report 2017–1039, doi:10.3133/ofr20171039.

---

## Author Response (AR1)

**Nat. Hazards Earth Syst. Sci. Discuss., https://doi.org/10.5194/nhess-2017-241**
Invited perspectives. A hydrological look to precipitation intensity duration thresholds for landslide initiation:
proposing hydro-meteorological thresholds
**Reply to Referee # 1**

We thank the reviewer for the detailed and constructive review. Below we address and reply to the
comments and questions. In Italic typesetting the original review is given, and in roman typesetting our
replies.

Thom Bogaard and Roberto Greco

**General comments**
*The paper offers a hydrological perspective of precipitation intensity-duration thresholds (hereafter, ID
thresholds) for landslide triggering, useful in early warning systems. The ID threshold is a well established
empirical model, as it is proposed in numerous studies. Several limitations affect these thresholds, as
summarized in this paper. The authors with this paper propose to move away from this "conventional" path
for future research, arguing that simple, even lumped, hydrological information should be introduced.
They propose a general framework, where thresholds should represent both landslide causes (dynamic
predisposing conditions) and landslide triggers. They argue that with ID thresholds only the latter are
(conceptually) considered. Hydrological information is related to the former, and should be represented by
something linked to soil water content.*
*I overall think that this is a good paper and well written. On the other hand, I also think that some
improvements can be made.*

We thank the reviewer for the detailed and constructive review. Actually, our conceptual approach is more
general than only focusing on shallow landslide (then, it would be indeed soil moisture). Our point is
that different hydrological information could be useful for landslide hazard assessment.

*In particular, two main issues the authors should better discuss are:*
*1. How to separate between landslide "causes" and landslide "triggers" in practice?*
*In other words: at which instant/timescale one should think that there is a switch fromcauses to triggers?*

This is an excellent point that we discussed at length ourselves. The reviewer is correct that categorizing
landslides based on "cause" and "trigger" requires a kind of time scale to separate the two. The discussion
on the timescale of trigger-cause is already half a century old (e.g. Sowers and Sowers, 1970). Wieczorek
(1996) defined triggering as an "external stimulus (…) that causes a near-immediate response in the form of
a landslide by rapidly increasing the stresses or by reducing the strength of slope materials.". In our own
advanced review WIREs Water (Bogaard-Greco, 2015) we summarized the trigger-cause as: "A trigger is
thelast push for a slope to become unstable, whereas thecause is the underlying, often long term, change
thatoccurred preparing the slope for failing.".
So we see the trigger as 'the last push' with near-immediate effect and consequently the hydrological cause
is all before that. We agree that when adapting our proposed framework, to use the cause-trigger concept
for defining regional landslide initiation thresholds, it becomes eminent to start defining timescale to
distinguish between trigger and cause.This will be different for different landslides and slopes.
We agree with the reviewer we did not discuss this and we will add a short discussion on the definition of
the timescale of the trigger event both in introduction and in conclusion section, as well as in the description
of the discussed examples, which point out how the timescales of both trigger and cause are strongly
related to the effective hydrological processes of a specific site.. However, for us, the more 'mathematical'
or 'precise' defining of the trigger timescale in the various situations is out scope of this invited perspective
and more for follow up work, when the community starts adapting the proposed concept.

*2. How to manage the higher modeling freedom (respect to PID thresholds) that one can introduce by
hydrological analyses?*
The reviewer is correct that looking for variables different from rainfall to define thresholds gives in principle
more freedom. However, the basic idea is that the choice of the most suitable variable should be guided not
only by pure statistical analysis (that is, how the threshold performs), but also, and mainly, by the
identification of the hydrological processes that, for the kind of landslide and geo-morphological context, are
expected to be responsible for "causing" the conditions predisposing to a landslide. The statistical analysis
could be regarded as a tool to confirm if the process identification is correct or not.

*More details on these two points are given in the specific comments (comments to L253 and L 262-264).*
*Finally, I recommend minor revisions for this manuscript.*

**Specific comments**

*L 20: "the conceptual idea is that precipitation information is a good proxy for bothmeteorological trigger and hydrological cause". It cannot be said that, in general, researchersderiving ID thresholds and their users have this conceptual idea in mind.This is a move of the authors which is not fully justified. So I think that this sentenceshould be rewritten, perhaps writing something on the fact that it is in general thoughtthat precipitation information can be linked by simple relationships to landslide occurrence,without explicitly taking into account hydrology.*

The reviewer is correct we of course cannot speak for all researchers/groups who derived ID thresholds that this was the conceptual idea. They also could have other justifications, like: it gives (statistically) good/useful results. We will rephrase to make clear it is our interpretation, and that the practical background of precipitation ID is that often only meteorological information is available when analyzing (non-) occurrence of shallow landslides, and that, at the same time, the conceptual interpretation of their success could be that precipitation is sometimes a good proxy for both meteorological trigger and hydrological cause.

*L 22: It is not fully clear what does "indistinct threshold" mean*

Agree. We will delete these words as indeed indistinct threshold is not defined. We will rephrase: "this approach suffers from many false positives …."

*L 36: "landslide is the most abundant hazard". Are the authors sure that "landsliding is the most abundant hazard"? Maybe say that it is "one of the most abundant naturalhazards", and add some references to literature (for instance: Sidle and Ochiai, 2013)Sidle, R. C. and Ochiai, H.: Landslides: Processes, Prediction, and Land Use, WaterResources Monograph, 2013.*

Agree, we will rephrase and add the reference for this:"one of the most abundant natural hazards"

*L 39 – 45: The three approaches listed by the authors are not all aimed to assess"landslide probability" in a strict sense (only number 3 is). In fact approach (1) leadsto an assessment of landslide "susceptibility", which is not exactly a probability, butan index of landslide proneness in a relative scale. Approach (2) does not provide ingeneral landslide probability, as most of the landslide triggering threshold schemes are"deterministic", and probability is in fact only in theory – but very seldom in practice– related to landslide triggering thresholds (Aleotti, 2004; Iiritano et al., 1998). Theauthors should clarify this point.*
*Aleotti, P.: A warning system for rainfall-induced shallow failures, Eng. Geol., 73, 247–265, 2004.*
*Iiritano, G., Versace, P., Sirangelo, B., 1998. Real time estimation of hazard for landslides triggered by rainfall. Environmental Geology 35 (2– 3), 175– 183.*

In a strict sense the reviewer is correct. We will replace 'probability' with 'possibility'. (and see comment L46 below).

*L 42: perhaps integrate literature on this, with other more recent papers (e.g. Peruccacci et al., 2017 and references therein)*
*Peruccacci, S., Brunetti, M. T., Gariano, S. L., Melillo, M., Rossi, M. and Guzzetti,F.: rainfall thresholds for possible landslide occurrence in Italy, Geomorphology, 290,39–57, doi:10.1016/j.geomorph.2017.03.031, 2017.*

Thanks for pointing to more recent literature.

*L 46: The term hazard may have a specific definition in the natural hazards field, related to the probability of the event to occur. So the authors should clarify that they refer to "hazard" in a broader sense. Perhaps in clarifying this they should cite a generally accepted definition of "landslide hazard". This comment is related to preceding one on L 39 - 45*

We are using the term (landslide) hazard in a broader sense, not in a probabilistic way: A (landslide) hazard is a natural phenomenon that might have a negative effect on people or the environment. We will clarify this in the text by adding a definition.

*L 63: the authors use both ID / PID when referring to precipitation intensity and durationthresholds. Only one way should be used*

Correct, we will use: (precipitation) ID

*L 63: "hazard" is perhaps not fully appropriate*

We will replace with: 'landslide hazard"

*L 70: add references to papers where a "probabilistic transition zone" is used*

We refer to Berti et al (2012) which is detailed on from L80 onwards

*L 88: It seems that authors are referring to works where antecedent precipitation isused (perhaps as a "measure of antecedent soil moisture content"). Here the authors should better clarify what they are referring to, and cite pertaining papers.*

We refer to measures indicating the wetness state of the soil anteceding a precipitation event triggering a landslide event. These are listed further on in the paper (L245 onwards). We will put some of those references here as well.

*L 88: It is unclear if antecedent precipitation should be seen in the authors' framework as an hydrological (cause) or meteorological (trigger) variable*

Hydrological cause: we detail on that from L245 onwards. We will add clarification here as well.

*L 90: again, here "hazard" is perhaps not fully appropriate*

Landslide hazard

*Figures 1 to 3: perhaps for a better comparison of the various curves it may be useful toplot in planes with the same axis range (e.g. x-axis of Fig. 1 goes from 0.1 to 100, whileFig. 2 from 0.1 to 1000). Also, it may be better that figures have the same appearance(e.g. no grid in the plot of Fig. 1; adjust font size in Fig. 3).*

Good suggestion, we will make the figures appear more homogeneous

*Figure 3: It is unclear how the dark grey area representing "landslide threshold" is derived from figure 2, as the area that it covers is narrower than that covered by thresholds in Fig. 2*

It is a generalised threshold summarizing figure 2. We on purpose narrowed the range for discussion reasons (like looking through eyelashes).

*L 171: It is unclear in which sense the ID threshold is "generalized"*

Broadly summarizing indication of the threshold.

*Figure 3: P is undefined (though its meaning can be easily understood from discussionin the text).*

As the reviewer suspects, P denotes precipitation, we will add this to the figure caption

*L 175: It is not clear why precipitation ID thresholds are "volumetric", as an infinitenumber of (I,D) or (H,D) pairs can be associated to a given event rainfall H.*

We mean to say: Every point on the threshold line depicts a volume, as H=I×D

*L 181: It is unclear why greater precipitation volumes should imply bigger landslides.Is this something reported in literature? I imagine that this is in general not true, asthe amount of rainfall gives little (or none) information on its spatial extension, and thusof that of the landslide. Also ID thresholds are derived using databases that usually report little information on landslide size, and to say that "the database consists for theoverwhelming majority of shallow landslides and debris flows" doesn't mean that the size of landslides is small.*

Sorry for the confusion, we are referring to deeper seated landslides: the amount of rainfall refers to unit surface area, so it cannot be related to landslide size, but it can be related to landslide depth. We will clarify.

*L 192 – 196: In this discussion the authors should mention that ID thresholds are sensitive to the way a rainfall event is defined, that is, mainly the maximum zero-precipitation interval within a rainfall event (See Vessia et al., 2014; Melillo et al; 2015). Cleary, the shorter this interval is, the shorter the length of rainfall*

*events will be. With long maximum dryness the events can be so long that different hydrological processes can takeplace. In this case rainfall events do not represent "the last push" but a mixture between "causes" and "triggers".*

*Vessia, G., Parise, M., Brunetti, M. T., Peruccacci, S., Rossi, M., Vennari, C. and Guzzetti, F.: Automated reconstruction of rainfall events responsible for shallow landslides, Nat. Hazards Earth Syst. Sci., 14(9), 2399–2408, doi:10.5194/nhess-14-2399-2014, 2014.*

*Melillo, M., Brunetti, M. T., Peruccacci, S., Gariano, S. L. and Guzzetti, F.: An algorithmfor the objective reconstruction of rainfall events responsible for landslides, Landslides, 12(2), 311–320, doi:10.1007/s10346-014-0471-3, 2015.*

This is our point. We do discuss this point in L161-L165. However, we agree with the reviewer that this point should be highlighted here as well. Thanks.

*L 253: The authors should discuss how to separate between the time scales of "causes" and those of the "triggers". In other words, how to switch, in practice, from the "cause" hydrological analysis (storage), to the "triggers" meteorological analysis (rainfall)? In other words, how does the framework the authors propose contribute in removing the subjectivity of identifying the rainfall that represents the "trigger"/"lastpush" (see comment on L 192 – 196)?*

See reply with general comments

*L 253: Another point is: hydrology may be in general important also during the "triggering" process, while in the authors' framework it is not explicitly taken into account. Arethe authors implicitly saying that "hydrology of the last push" can be taken into account without a significant processing of rainfall data?*

The point we want to make here is that we propose that an effort is made to separate longer time scale causes with shorter time scale triggers. Of course we do not imply other mechanisms cannot take place, but processes evolving over shorter scales are more directly related to the characteristics of rainfall event (i.e. the intensity), while rainfall effects on long-term processes are smoothed.

*L 262-264: "However there are several possible choices of hydrological variables to be plotted along the cause-axis, such as soil water content, catchment storage, representative regional groundwater level and similar". This implicitly reveals that a high degree of subjectivity follows from the framework that the authors propose. Researchers do generally agree that subjectivity of the ID threshold assessment is significant, in spite of his simplicity. For instance, one source of subjectivity in ID thresholds is related to the choice of the maximum zero-precipitation interval to define rainfall events (seecomment on L 192-196). This is known to impair comparisons between thresholds, which thus makes it difficult to search for general landslide triggering thresholds. The framework that the authors propose seems to possibly bring a higher heterogeneity of the analyses, and thus maybe can in practice represent a step backwards for finding unifying concepts. By introducing hydrological analysis, researchers may have more freedom in choosing models and parameters for estimating the "cause" variable (antecedent soil water content). This may represent a possible way to manipulate theresults so that the performances of the resulting hydro-meteorological thresholds appear to be higher than they actually are. Thus, the authors should discuss how one can prevent this, perhaps by highlighting the importance of always performing validation analyses, i.e. to test developed thresholds against a sub-dataset which is not used in calibration.*

The point we make is that (hopefully) more process knowledge will be in the graphs and less statistics. Indeed there will be a higher degree of freedom from a statistical point of view, but we argue also higher causality. The reviewer is right that this subjectivity can be a problem, but we take the stand that searching for a hydrological cause will improve the physics behind the thresholds whereas now, we rely only on statistics.

*L 319: "ID thresholds neglect the role of the hydrological processes" is a strong statement. Indeed it may be written that hydrological processes are too simplistically represented by ID thresholds. In other words, precipitation is the main cause of landslides, but the main problem is: how to process precipitation information to obtain thresholds that perform well in forecasting landslides? And, of course, ID thresholds certainly do not represent the best way to processes rainfall data.*

We do not agree with the reviewer that this statement is too strong. There is no attempt to put hydrology into the threshold and only in case of relatively linear relationships between hydrology and precipitation this type of thresholds would hold. But hydrology (pore pressure built-up in the subsurface), is not linearly related to precipitation. So, yes, we can look at ID thresholds in terms of statistics, but we argue one could also look it from the point of "being right for the right reason".

*L 332: I agree that one downside of spatially-distributed physically based models is that they require a "well calibration". However to estimate catchment storage (as in Ciavolella et al., 2016), requires a well calibrated model too. The authors should discussbetter this point.*

Point well taken. Indeed, in practice also some of calibration will remain required, although to a lesser extend then calibration of a fully distributed model. We will add this in the discussion.

*L 233: A sketch explaining the approach the authors propose can be useful for readers.*
Good suggestion, we will try.

**Technical corrections**

We are thankful for these technical corrections. We will correct them in the revised version
*L 42: Caine instead of Cain*
*L 45: maybe something is missing as citations finish with a ";"*
*L 71: "separation" instead of "separator"*
*L 78: remove "," after "conditions"*
*L 142: perhaps replace "for regions or areas not pertaining to this area" with "otherregions or areas"*
*L 147: "threshold" instead of "thresholds"*
*L 197: perhaps "phenomena" instead of "hazards"*
*L 198: "related" instead of "relate"*
*L 223: "thresholds" instead of "threshold"*
*L 255: perhaps "field" instead of "terrain"*
*L 283: "specific" instead of "particular"*
*L 318: "limitations" instead of "limitation"*
*L 326: "interpretations" instead of "interpretation"*
*L 331: "physically based" instead of "physically-based"*

**Nat. Hazards Earth Syst. Sci. Discuss., https://doi.org/10.5194/nhess-2017-241**
Invited perspectives. A hydrological look to precipitation intensity duration thresholds for landslide initiation: proposing hydro-meteorological thresholds
**Reply to Referee # 2; Roy Sidle**

We thank Roy Sidle for the detailed and constructive review. Also his insights in original literature is very valuable. Thanks so much. Below we address and reply to the comments and questions. In Italic typesetting the original review is given, and in roman typesetting our replies.

Thom Bogaard and Roberto Greco

*This paper offers a refreshing and needed critique of the ID relationships commonly used in regional (and global) landslide predictions. Furthermore, it proposes an improved approach that includes metrics of predisposing 'causes' and 'triggers' of shallow landslides. As such, it should stimulate new research in this arena that will benefit landslide and hillslope debris flow prediction. It will be a valuable contribution to NHESS with some moderate revision.*

*I noted several times in my review that follows that ID relations (at least some previously published ones) have erroneously reported data as 'individual storms', which were obviously not individual events (i.e., very, very long durations). Additionally, on the opposite side of the ID 'x-axis' there are instances of very, very short storms of high intensity triggering landslides – these appear to be bursts of intensity on saturated soils as noted by the authors or they could in fact represent a totally different process, like channel bed mobilization causing a debris flow. My recollection of reading through some older reports in which data were used to develop ID thresholds is that in some cases the described mass failure was more of a within channel debris flow. Off the top of my head, I am thinking of some of Rapp's early papers that were included in Caine's threshold. In any event, these anomalies should be considered or mentioned herein.*

The reviewer clearly indicates the importance of rainfall variation over a rain period. This indeed is extremely important as also discussed in our paper (L163-L166; L327). However, we agree, we should discuss this part more. We will add a new paragraph around L163 in which we detail on the effect of a) the uncertainty arising from precipitation measurements using gauges and b) the effect of using average intensities in the ID thresholds and that when moving from short to very long events it is not possible to have identical definitions of what an event is.

*I have noted a number of editorial suggestions directly on the manuscript which I will attach for the authors.*

Thanks, we are thankful for these and will benefit from them.

*More scientific technical comments are noted as follows:*

*Title: I would say "Hydrological perspectives on precipitation intensity – duration thresholds. . ..."*

The title starting with "Invited perspectives:" is a format of NHESS. We will discuss the title based on your suggestion (and of Ben Mirus) with the executive editors of NHESS.

*Lines 28-29: reword – "discuss" based on "associated discussion"*

OK

*Lines 86-88: yes, we tried this in our 1985 Hillslope Stability and Land Use book using antecedent rainfall information, but the problem was the lack of documentation of such antecedent rainfall data in earlier studies. Overall, we felt that it did improve the ID thresholds (at least conceptually).*

Thanks for sharing, we will make appropriate reference to it.

*Line 109: The term 'stormwater management' implies to me more of an urban planning context; that may be my bias, but you may want to add 'flood prediction' (or something like this) as well.*

We indeed use the term for urban stormwater management in this context. We will add the flood prediction.

*Line 125-126: I think that this is a key difference between practical applications of IDF and ID curves; that is most (or at least many) shallow landslides respond to sort-term intensity bursts which are not articulated in typical IDF's. You may want to mention this.*

Agree, indeed the rainfall durations considered for the derivation of IDF curves have nothing to do with the beginning or with the end of a rainfall event: they are time intervals of given duration during which some rainfall fell. And then, extreme observed values are considered (e.g. annual maxima). Shallow landslide tend to respond indeed to short term high intensity of rainfall which are often not visible anymore when averaging over larger time steps. We will include this aspect in our paper (see also below)

*Line 144: Try not to start sentences with "Figure x shows. . ."; this can be seen in the Figure and caption. Just directly say what you wish to say about the data in the figure and cite the figure in parenthesis at the end of the sentence.*

We will reformulate this.

*Lines 155-157: rework this sentence – understandable, but a bit confusing. Maybe just put 'mostly debris flow and some shallow landslides' in parenthesis. Furthermore, I think there are some issues with such very short 'landslide producing storms' reported in the literature that are captured in these cited thresholds. As you note, they are probably mostly debris flows, and upon inspection of some earlier papers that reported such short-term events, it seemed that the authors were referring to possibly a different process – e.g., debris flows caused by channel bed mobilisation. I looking into this matter in our 2006 landslide book, we actually threw out some of these short-term rain events when constructing new ID curves because we were convinced that they represented different triggering processes.*

We rephrase as follows: "For landslides triggered by short precipitation events (D ≤ 1 hr), the slopes of the IDF and ID curves substantially coincide (Figure 3). ".

Furthermore we will add it after line 161: "This is counter-intuitive, as during long-lasting wet periods landslides are usually more frequent, while many debris-flows triggered by very short and intense storm originate from channel bed mobilization rather than being (new) mass movements"

*Lines 158-166: I agree that this is problematic, and I feel (as you state) that ignoring short-term peaks of rainfall in an otherwise long-duration, lower intensity event is the main reason for this problem. Based on my work and that of others, I always say that one common scenario for shallow rapid landslide initiation is a long storm of low to moderate intensity, with a peak intensity occurring near the end of the event. Another issue here, I agree that the longer return periods for landslides triggered by long-duration, low intensity storms is counterintuitive; however, when we looked into the actual data for some of these so-called long duration events that triggered landslides (in reviewing references for the 2006 AGU landslide book), it became apparent that some of the data included in these ID relationships were not strictly 'individual events', rather these were based on a longer period of rainfall leading up to the landslide. – thus, a direct comparison with some of these so-called long-duration landslide triggering 'events' with IDF curves for actual individual events may be a bit problematic. You probably should mention this potential discrepancy. My point is probably only relevant for the very long 'events', but it may be worth mentioning.*

(See also reply after general comment at top and after Line 125-126) The importance of rainfall intensity variation should be elaborated on, and as stated before, we will add a paragraph discussing this in more detail.

*Lines 185-188: This sentence is a bit confusing; it seems that you are referring to reported data when you saw the 'vast majority of empirical thresholds fall between . . .". Are you saying that for other studies most of the landslide reported would fall between thresholds of 10 to 100 mm? If so, you need to cite some references. But I am not sure that is what you are trying to say here. Anyway, please clarify. (and you overuse the expression 'vast majority' – just say most or the majority).*

We reformulate: "Many of the reported empirical precipitation thresholds has between 10 and 100 mm of accumulated precipitation. However, also <10 mm and >1000 mm volumes needed for landslide initiation have been reported."

*Lines 194-196 See my previous comment about data for very long duration 'events' that are likely not individual events.*

*Lines 209-210: In addition to my comment in the text, also see my previous comment about data for very long duration 'events' that are likely not individual events.*

Agreed. See response above

*Lines 227-231: Very complex sentence and a bit awkward. Can you rewrite this or try to break it up a bit?*

We reformulate: "Hence, it was possible to define non-dimensional variables comparing the meteorological triggers with the infiltration and storage capacity of the soil cover. This non-dimensional hydro-meteorological threshold performed slightly better than the precipitation ID threshold in separating events resulting in factors of safety smaller and greater than 1.3."

*Lines 240-241: I don't mean to be beating a 'dead horse' again, but such long events are obviously not 'events'; they were probably included in databases because this was the only precipitation record reported.*

Agreed. See response above

*Line 257: Why do you say 'was preferred'? by who?*

We intended to express that in several studies another approach was followed, that of distributed physically-based modelling. We will change in: "This is a largely unexplored terrain, although we recognize that data availability can be cumbersome. " The discussion on the option of using physically-based distributed models is done elsewhere in the paper (In introduction and in conclusion L330).

*Lines 258-260: Reword the first sentence to note that the trigger axis refers to the rainfall characteristics (intensity) responsible for initiating the landslide. When you say "depends on the local situation" – I think you mean both available data and the rainfall characteristics that are responsible for landslide initiation in that area.*

Reformulated as: "Concerning the 'trigger'-axis, there is little debate; it is the rainfall intensity responsible for the short-term last push initiating a landslide."

*Line 276: What do you mean by 'discharge intensity'? This is a rather unconventional term.*

Reichenbach et al (1998) used: event intensity (in $m^3sec^{-1}km^{-2}$). We will explain and use "specific discharge".

*Line 282: What is low/high storage?*

We mean to say how much water is stored in a catchment compared to its maximum storage capacity. We clarify.

*Lines 286-287 (and the sentences that follow): I think this phenomena occurs for deep seated landslides like earthflows of slump-earthflows – maybe better to state this to avoid confusion, because you are mostly focussing on shallow landslides. There is some older work in Japan that has clearly showed such relationships with earthflow reactivation and a threshold groundwater depth. I believe mark Reid also published a paper on this from earlier work in Hawaii.*

Thanks, indeed this mainly holds for deeper seated landslides. Thanks for the reference. We will rephrase: "In some cases, mainly deeper seated landslides, ….."

*Line 322: Again, not all data in these ID relationships were for individual events.*

Agree

*Line 323: you mean even when they are developed for small areas?*

Yes. Thanks

*Line 338: These will be particularly valuable in developing countries.*

True!

*I really like the message in the last paragraph of the Conclusions! Well articulated.*

Thank you

**Nat. Hazards Earth Syst. Sci. Discuss., https://doi.org/10.5194/nhess-2017-241**
Invited perspectives. A hydrological look to precipitation intensity duration thresholds for landslide initiation: proposing hydro-meteorological thresholds

**Reply to Referee # 3 Francesco Marra**

We thank Francesco Marra for the insightful comment. Below we address and reply to the comments and questions. In Italic typesetting the original review is given, and in roman typesetting our replies.

Thom Bogaard and Roberto Greco

*The authors analyze the concept of precipitation intensity-duration (ID) thresholds for shallow landslides and debris flows from a hydro-meteorological perspective to propose a new approach to the problem. The contribution is largely welcome since it suggests new approaches and perspectives to overcome important, but sometimes neglected, limitations of the commonly used methods.*

*Within the interesting analysis of the relationship between ID thresholds and IDF curves, the discussion so far neglects the impact of rainfall estimation uncertainty and its possible dependence on rainfall duration and return period (e.g., Krajewski et al., 2003, www.dx.doi.org/10.1623/hysj.48.2.151.44694; Ciach and Krajewski, 2006, www.dx.doi.org/10.1016/j.advwatres.2005.11.003). Recently, rainfall estimation uncertainty caused by the use of rain gauge measurements (still the most common source of rainfall estimates in this field) was shown to significantly affect the derived ID thresholds causing systematic bias (Nikolopoulos et al., 2014, www.dx.doi.org/10.1016/j.geomorph.2014.06.015). This systematic bias is caused by (i) systematic rainfall patterns observed around the triggering locations and (ii) the use of log-transformations within the derivation of the ID (Marra et al., 2016, www.dx.doi.org/10.1016/j.jhydrol.2015.10.010). At least for durations 2 days, these rainfall patterns were observed to be related to the return period of the triggering rainfall (Destro et al., 2017, www.dx.doi.org/10.1016/j.geomorph.2016.11.019). Consequently, the 'slope' of the ID threshold is affected by rain gauge sampling, and this potentially undermines the comparison between the slope of ID thresholds and IDF curves reported in the manuscript. In particular, I think this aspect should be discussed when the authors say [line 157 and following]: "On the other hand, for longer precipitation durations, ID thresholds have smaller slopes than IDF curves.This means that landslide initiation on the right side of the graph (lower precipitation intensity with longer duration) would occur with rapidly increasing return periods of precipitation events". In fact, the observed pattern could be caused/emphasized by the sampling issues discussed by Marra et al., (2016) and Destro et al. (2017).*

*To conclude, IDF curves are expected to vary within the examined region (generally a regional, if not global, scale); it is thus unclear to me how the idealized IDF curves in Fig. 3 have been drawn. The 'slope' of the curves (i.e. the dependence of I with D) at the regional scale may change so that the curves should be better represented as a shaded area – such as done for the ID thresholds.*

*This was a technical comment on an introductory aspect of the study; this being said, I repeat my compliments to the authors for the manuscript and the new proposed perspective.*

*With kind regards, Francesco Marra*

Dear Francesco Marra

Thank you very much for you excellent comment. We strongly focus on the hydrology whereas you bring up the point of the effect rain patterns and rain measurements (rain gauge based precipitation observation, spatial organisation of rain and associated uncertainty) that so far we on purpose have left out of the article. We tried to keep the article brief as is requested for an "invited commentary". Our focus is that only using precipitation data for landslide hazard assessment is neglecting the importance of hydrological processes in landslide initiation. Hereto, we make a quite extensive 'problem' analysis showing the currently used ID thresholds have limited physical meaning, at the best they are statistically interesting to use. However, you are of course fully right that precipitation estimates and uncertainty can also be responsible for (part of) the observed low slopes of the ID thresholds. So, in hindsight we agree this is a too important aspect not to address, so we will add a short discussion on the effect of precipitation measurement uncertainty as depicted in your comment.

Your second point is about the (slope of the) IDF curve that we published in Figure 3. Yes, this is a somewhat arbitrary but representative, set of IDF curves (that is, the chosen slope is not far from those of the examples of IDF curves of Fig. 1, which refer to very different locations around the world). Like we write in the caption "Schematic" and in our opinion useful for the argumentation. It is however a good suggestion to show, also graphically, in Figure 3, that the IDF are schematic, so we will adapt the figure and explain it in the text as well.

**Nat. Hazards Earth Syst. Sci. Discuss., https://doi.org/10.5194/nhess-2017-241**
Invited perspectives. A hydrological look to precipitation intensity duration thresholds for landslide initiation: proposing hydro-meteorological thresholds

**Reply to Referee # 4;** *B. Mirus,* bbmirus@usgs.gov

We thank Ben Mirus for this pro-active, stimulating and detailed review of our discussion paper. Below we address and reply to the comments and questions. In Italic typesetting the original review is given, and in roman typesetting our replies.

Thom Bogaard and Roberto Greco

*The authors present a much needed discussion about some systematic problems with precipitation intensity-duration (ID) threshold approaches for predicting shallow landslides.In particular, they point out an unfortunate lack of reasonable constraints on the max/min duration of rainfall events, and also discuss how these unbounded events can affect the average intensity and predictive capabilities. Both issues are largely ignored in many studies focused on developing and testing ID thresholds, so it's a worthy discussion about some crucial sources of error. The authors also highlight the potential importance of hydrological information in addition to precipitation characteristics, which have not been systematically incorporated into landslide early warning criteria. In my opinion the most innovative contribution presented in the manuscript is the comparison of the rainfall intensity recurrence intervals and ID thresholds for landslide initiation from the literature, along with the contour lines of cumulative storm totals (Fig. 3). This is a new and intuitive way to broadly illustrate their point about some problems with the ID threshold concept.*

We agree that Figure 3 is key in our argumentation of the issues linked to precipitation ID thresholds.

*However, my primary concerns with the manuscript are twofold:*
> *(1) Limited concrete guidance is provided on how to apply the proposed "cause-trigger" framework, so the potential novelty of the approach seems somewhat overstated.The general concept that both the predisposing factors (e.g. antecedent wetness) anda rainfall triggering event are needed to explain shallow landslide initiation is alreadygenerally accepted and has in fact been implemented in a number of landslide initiationthresholds. For example, two different rainfall thresholds developed for the Seattle area explicitly account for antecedent factors: (a) the recent-antecedent cumulative precipitation threshold compares the 3-day triggering rainfall to the 15-day antecedent rainfall (Chleborad et al, 2008), and (b) the Antecedent Water Index is used with an exponential ID threshold for events between 10min and 10days in duration, thoughstorms are generally less than 24hours (Godt et al., 2006). The authors also cite several other papers (including some of their own published work) that in various ways incorporate antecedent wetness as a measure for the predisposing factors prior to the triggering rainfall event using soil water balance modeling or catchment storage. As such it is not clear how the "cause-trigger" approach is truly novel, but rather seems to be a new term for a topic in need of further exploration. (As an aside, the term"predisposing factors" seems to be a more appropriate term: without context "cause" could be misleading since both predisposing factors and a triggering event are needed to cause a landslide.)*

The objectives of this invited perspective are to: (a) critically analyse the precipitation ID thresholds for shallow landslides and debris flows from a hydro-meteorological point of view; and (b) propose a conceptual framework for lumped hydro-meteorological hazard assessment based on the concepts of trigger and cause.
We think we do not claim to be 'truly novel' in our paper. Obviously, as also discussed in the paper at length, we write this perspective-paper standing on shoulders of giants; the many colleagues who have shed light on this topic before us. In our article, we brought it into an overarching conceptual framework under which new research could (should) be done (L340-L350), also because many of the existing examples - like the two you point out - look for the causal relationship between antecedent precipitation and landslide occurrence merely by maximizing some measure of correlation, while our proposed framework invites to look at the most relevant hydrological process for the considered context. Only in the abstract (L27) we did write "novel trigger cause concept". Here we will reformulate in "we propose a trigger-cause conceptual framework"

The framing into 'trigger-cause' and then especially the "cause" is indeed debatable We defined the word "(hydrological) cause" as the predisposing (hydrological) condition of an area under study (L260, L320). We will define our use of the word "cause" earlier in the paper (section 3).

> *(2)* *The critical issue of data availability is understated.The topic of data availability is largely avoided until the very end of the conclusions, at which point it comes across as an afterthought instead of the main reason the precipitation ID threshold has been employed successfully for decades. Without continuous records of appropriate data during historic landsliding events it is challenging (if not impossible) to develop and test alternatives to the precipitation ID threshold. The reality is that rainfall data is widely available and has been for some time, which has facilitated useful, albeit somewhat flawed tools for assessing landslide potential for a number oflandslide-prone areas. Secondly, rainfall can be predicted in advance with considerable accuracy, so despite some errors in ID thresholds, the trade-off between appropriate lead-time using weather forecasts and threshold accuracy must at least be considered when arguing for alternative threshold approaches. Without a more balanced discussion of data availability it's not entirely clear whether the authors are arguing for better analysis of rainfall data that distinguished between the "causing" rainfall and the "triggering" rainfall or if the authors suggest that rainfall is not an appropriate data source for the "cause" variable and the ID threshold concept has been employed incorrectly for very long and very short duration storms. Although the Invited Perspective highlights both these problems with the ID threshold approach, it remains unclear how the"cause-trigger" approach can be used to solve these problems within the context oflimited data availability.*

Indeed, we decided to first set-up the problem, then the concept and lastly, the "afterthoughts". However, we do mention the issue of data availability several times (e.g. L255, relatively in the beginning of section 3). We do agree that the data availability is the issue (at the moment) and we agree to stress that in our paper also in writing why the traditional ID thresholds are often preferred/used. May we add that once the driving hydrological process is identified, modelling can supplement the lack of data, while this is impossible with rainfall alone?
The fact that rainfall data is widely available is, in our opinion, maybe one of the reasons for misuse of ID thresholds. I our opinion "somewhat flawed" seems an understatement. Could it not be that relying on rainfall records prevented us from digging deeper? Secondly, many papers address the issue of how representative this rainfall information is. By 'uncritically' linking landslide occurrence to the nearest precipitation record, we seem to prefer practical/statistical correlation over causal relation. In this perspective, this is one of the two aims.

> *After addressing these two issues regarding the novelty of the proposed approach and the availability of data for landslide initiation thresholds, the authors' hydrologic perspectives on the precipitation ID approach for landslide prediction will be a valuable contribution and will surely be of considerable interest to readers of NHESS. I suggest a number of general and specific revisions prior to publication, outlined in the sections below. Thank you for your consideration.*
> *Kind regards, Ben Mirus*

> *General Revisions:*
> *(a) Explain how the details of how the "cause-trigger" concept can be distinguished from prior contributions that consider antecedent conditions, or qualify the novelty ofthe proposed approach within the context of such prior work.*
> *(b) Provide more concrete guidance on how the "cause-trigger" framework could beapplied for future studies. In particular, how should researchers constrain the durationof storms to distinguish between "cause/predisposing factors" and "trigger"?*
> *(c) Include a more balanced discussion of what data could reasonably be obtained toinform the "cause" axis for any landslide early-warning threshold relative to the widelyavailable (and forecastable) input of rainfall.*

See above

> *Specific Edits:*
> *L1: I agree with the revision to the title suggested by Roy Sidle and would further suggest removing the second phrase since there no specific hydro-meteorological thresholds are proposed. Thus the suggested title is shorter and more precise: "Hydrological Perspectives on the Precipitation Intensity Duration Thresholds for Shallow Landslide Initiation"*

The format of the title is set by NHESS. We will discuss this with the editor in charge. We agree: Hydrological perspectives etc is to-the-point.

*L13: Provide some citation or definitive evidence for the strong, yet disputable statements like "vast majority" and "never" . . . otherwise such pronouncements should be avoided in scientific writing. Furthermore, as is later argued in the manuscript, precipitation does not actually initiate the landslide. Suggest revising to: "Many shallow landslides and debris flows are rainfall induced."*
The intention of an invited perspective is also to raise discussion and have sharp edges. We therefore allowed ourselves a writing style which is "less scientific" as generally seen in scientific articles. But as all reviewers bring this point on, we clearly 'overdone' it and we will discard such strong pronouncements.

*L22: What does "indistinct" mean? Thresholds are by nature distinct. On the other hand, the errors resulting from application of distinct thresholds over broad areas reflects the heterogeneity of natural systems. In theory, each hillslope/hollow has a unique threshold that must be averaged over some area and some time to create auseful tool for landslide early warning.*
Agree. We will delete these words as indeed indistinct threshold is not defined. We will rephrase: "this approach suffers from many false positives …."

*L27: Again, calling this a novel conceptual framework is an overstatement. See general comments.*
See reply above. Will change in: A conceptual framework

*L36: References to support this claim? I was not aware landsliding is the MOST abundant hazard. At the very least it should be qualified as a natural hazard, sincemany health or other hazards could be considered more abundant and/or detriment also socio-economics.*

Agree, we will rephrase and add reference for this: "one of the most abundant natural hazards"

*L40-43: Unclear from the description provided here how 1) and 3) are different when applied to assessing landslide probability. Perhaps some example citations later in the paragraph could help distinguish between the two.*
See also Reviewer #1: We will replace 'probability' with 'possibility'

*L59: I recommend also citing Anagnostopoulos et al. 2015 when discussing model complexity.*
Thanks for the suggestion.

*L74-75: Perhaps include some more recent citations that are less than 10 years old?*
You have a point here, we will add more recent references.

*L78-79: Yes. Also there is considerable error introduced by the heterogeneity that must be "averaged out" for a PID threshold to be developed over an area of interest.*
Yes

*L86-88: Are you proposing something that is better than soil moisture? It seems that soil moisture would be better than the other variables suggested (albeit harder to measure),so it is seems counterintuitive to state that these studies are "limited" to measuresof antecedent moisture content.*
Correct, "limited" should be removed: "however they were mainly including measures of antecedent soil moisture content, which may not represent the most suitable variable for any kind of landslide"

*L88: Never say never. In general it is unwise use this word in scientific writing unless it can be rigorously confirmed, which is almost "never" possible. Suggest revising to"not" or "have not been the subject of"*
As replied before we have been somewhat over-enthusiastic in our writing style. We agree to reduce the use of the strong pronouncements.

*L106: Here and elsewhere the abbreviation switches from PID to ID. . . either is fine,but use only one consistently throughout.*
Right, we will use ID

*L149: What is an "absolute" value of a threshold? Do you mean, for example the xandy-intercept values? Revise for clarity.*
*L152: Again, what is the "absolute" precipitation ID?*
We mean unscaled measured precipitation values

*L153: At some point in this part of the discussion you should mention the novel use of duration-frequency curves by Fusco et al., 2017. They use this concept to examine temporal and spatial patterns of pressure head states that predispose slopes to recharge and/or landslide initiation. I think it is a nice example of how the "cause" concept you propose could be implemented practically.*

Good suggestion. Thanks for pointing us to it. This is indeed a nice example.

*L167: Yes. This also leads to questions about how storm durations are defined, particularly since longer storms are more likely to include actual breaks in precipitation where drainage and ET can be more effective in reducing landslide initiation potential.*

*L181-184: Revise these sentences for greater clarity. It seems like the main point is that if larger cumulative precipitations are needed to initiate landslides at lower intensities we would expect that to be reflected by the larger landslides in the inventory, but the inventory you reference is mostly small landslides, so an alternate interpretation is that the slope drains while it's raining? At least the last sentence is incomplete to communicate the message more clearly.*
Sorry for the confusion. The amount of rainfall refers to unit surface area, so it cannot be related to landslide size, but it can be related to landslide depth. Indeed, we are referring to deeper seated landslides. We reformulate.

*L186: Technically this (between <10 and >1000) is not a range, it is unbounded. Doyou mean between >10 and <1000? Revise for accuracy.*
We reformulate: "Many of the reported empirical precipitation thresholds has between 10 and 100 mm of accumulated precipitation. However, also <10 mm and >1000 mm volumes needed for landslide initiation have been reported."

*L187: Again, such strong statements like "vast majority" should be supported by a number of independent citations or other evidence. Otherwise avoid this term.*
Agree

*L200: Not sure this is the most appropriate phrasing. The real utility of ID thresholds is that they are not at all cumbersome to use, but rather involve a very simple and easy interpretation: does the rainfall intensity and duration plot above or below the threshold line? Maybe more important point is that ID thresholds applied locally need a "calibrated range" for storm duration whereas regionally and globally they are misleading since there is too much spatial variability in rainfall and hillslope hydrologic responses for accurate predictions.*
Good point, the ID thresholds are maybe too easy to use: We change the word "use" into "interpretation". This makes the interpretation of ID thresholds cumbersome.

*L206: Napolitano et al., 2015 is another good reference to include here as they also used seasonal variations in antecedent soil wetness to identify different thresholds forwinter vs. summer.*
Thanks for the suggestion. We would also add some remark about the fact that, when this kind of thresholds accounting for previous precipitation (seasonal variation) have been proposed, then the considered antecedent duration is usually the mere result of a correlation analysis. By exploiting process knowledge, instead, you directly can look at the most appropriate physical cause-effect relationship (and thus variable).

*L218-219: Indeed, this is the concept underlying the recent-antecedent cumulative rainfall threshold of Chleborad et al., 2008, except they use prescribed durations, which have since been statistically tested with receiver operator characteristics (Scheevel etal., 2017)*
Thanks for this insight. We add this.

*L229-231: The wording of this sentence is confusing. What is the significance of this separation between near-failure events at FS < 1.3? Without reading the papers listed before it's not really clear why this is relevant.*
We reformulate: "Hence, it was possible to define non-dimensional variables comparing the meteorological triggers with the infiltration and storage capacity of the soil cover. This non-dimensional hydro-meteorological threshold performed slightly better than the precipitation ID threshold in separating events resulting in factors of safety smaller and greater than 1.3. The choice of referring to a factor of safety larger than 1.0 was dictated by the actually observed soil conditions during the monitoring period"

*L247: Not exactly, for such studies soil water content is not usually measured directly.Suggest revising to "proxy" instead of "measure".*
Good suggestion. We will use "proxy"

*L261: Again, I much prefer the term "predisposing condition" (or factors) over "cause" since both the trigger and predisposing factors essentially conspire to "cause" the increased pore pressures and reduced strength that initiates a landslide.*

Although we see your point we prefer the word "cause" (defined as "predisposing (hydrological) condition"), because it is a way to highlight that we are looking for the identification of the causal hydrological process.

*L263: Although perhaps beyond the scope of this paper, the "cause-triggers" approach ought to be more universally applicable to landslides, including those triggered by earthquakes or erosion. For example, an earthquake may trigger more landslides in wet vs. dry soils. Similarly, erosion at the toe of a slope may allow it to become more predisposed to failure during a rainfall event. Perhaps worth considering as you explore and develop this framework in the future.*
Yes, this is indeed an outlook we have. And yes, your examples are excellent. I think this is a very good suggestion for future work.

*L264: What kind of "storage" is this? Do you mean how much water is stored in the catchment (e.g. an effective saturation)? Or do you mean how much water the catchment can store (i.e. storage capacity)?*
We mean to say how much water is stored in a catchment compared to it maximum storage capacity. We clarify

*L286: How is a catchment itself more or less permeable? Bedrock permeability is clearly a measure you could consider, but very permeable bedrock would result in all groundwater recharge and no runoff (or landslides?). Perhaps more relevant would be the thickness and hydraulic conductivity of the soil, which ultimately are reflected byhow quickly the catchment drains.*
Indeed not well formulated, it indeed links to soil hydraulic characteristics. Chitu et al found that two catchment with low infiltration capacity and thus larger direct runoff fraction links to landslides whereas in the other catchment the underlying hydrological process was infiltration and pore pressure build-up. We will reformulate.

*L290-291: There are a lot of things mixed up in here, which makes it difficult to relate to the primary topics of the article. First, this is not a shallow landslide, so perhaps this is a bit of a tangential argument for this paper, but it seems that the main point is mobility for a large, slow moving landslide can be related to groundwater levels. That's fine.However, it's not clear groundwater levels would be a good proxy for conditions favouring shallow landslides, particularly since deeper groundwater levels might not respond until after shallow soils on hillslopes have drained and are no longer susceptible to failure.*
Yes, you are completely right here, this example is not a shallow landslide, but indeed a deeper-seated, reactivating, coastal bluff type of landslide. This will be explicitly mentioned, and yes, regional GW does not need to be a good proxy for shallow landslides but we also did not claim that either. But we agree this was not clear. Thanks for pointing to this.

*L310-314: OK. This makes sense, but can you provide more concrete guidance or framework for evaluating the appropriate "cause" variable? Also, can you provide some balanced perspective of how readily available those types of data may be relative to rainfall? An example Figure 4 might be helpful.*
In this perspective we aim to give direction to future research by providing a problem analysis and an overarching framework. Cautiously, we also give some examples, and we later on discuss that the approach will (currently) suffer from data availability. In my own experience, we ran into this problem as well when searching for hydrological information to use. However with more and more data coming available, we argue the community could try to make the step from "statistical/practical' threshold to "causal relationships".

*L322-323: This is a somewhat subjective (i.e. value) judgment, which is tangential to the discussion presented here. The perceived or tangible value of predicting even 1/100 landslide events correctly at the expense of many false alarms is an entirely different question. Probably "predictive accuracy" would be more appropriate.*
Thanks for the suggestion. We will change into "predictive accuracy"

*L340-349: I completely agree with these statements and don't wish to argue with the sentiment, but at the same time the conclusions are rather wordy and not particularly satisfying or informative. Another (shorter) way of saying this is that hydrologic information could improve individual thresholds for shallow landslide initiation, but the type of hydrologic information that is most appropriate will vary based on location and data availability. So then how do we go about addressing this issue?*
Indeed the last paragraph is somewhat wordy. One reviewer likes it a lot, the other one a bit less. We will try to use shorter sentences in the last paragraph to increase readability.

*L350: The last-minute mention of remote sensing comes across as a bit of an afterthought and it is not clear how this very broad suite of information products can be used to constrain hillslope water balance. Why not soil moisture monitoring?*

Indeed is the explicit mentioning of RS unnecessary and an "afterthought". We will replace by mentioning the increasing hydrological data that become available (without specifying).

*References Cited:*
*Anagnostopoulos, G.G., S. Fatichi, P. Burlando, 2015, An advanced process-based distributed model for the investigation of rainfall-induced landslides: The effect of process representation and boundary conditions, Water Resources Research,doi:10.1002/2015WR016909*
*Chleborad, A.F., R.L. Baum, J.W. Godt, P.S. Powers, 2008, A prototype for forecasting landslides in the Seattle, Washington, Area, Reviews in Engineering Geology, doi:10.1130/2008.4020(06).*
*Fusco, F., V. Allocca, and P. De Vita, 2017, Hydro-geomorphological modelling of ashfall pyroclastic soils for debris flow initiation and groundwater recharge in Campania(southern Italy), Catena, doi:10.1016/j.catena.2017.07.010.*
*Godt, J.W., R. L. Baum, A.F. Chleborad, 2006, Rainfall characteristics for shallow landsliding in Seattle, Washington, USA, Earth Surface Processes and Landforms, doi:10.1002/esp.1237.*
*Napolitano, E., F. Fusco, R. L. Baum, J.W. Godt, P. De Vita. 2015, Effect of antecedent hydrological conditions on rainfall triggering of debris flows in ash-fall pyroclastic mantled slopes of Campania (southern Italy), Landslides, doi:10.1007/s10346-015-0647-5)*
*Scheevel, C.R., R.L. Baum, B.B. Mirus, J.B. Smith, 2017, Precipitation thresholds for landslide occurrence near Seattle, Mukilteo, and Everett, Washington: U.S. GeologicalSurvey Open-File Report 2017–1039, doi:10.3133/ofr20171039.*

Hydrological perspectives.  on precipitation intensity duration thresholds for landslide initiation: proposing hydro- meteorological thresholds

Thom.Bogaard[1] and Roberto Greco[2]

[revised manuscript text omitted]

However, several shortcomings are frequently recognized and discussed. For example, Berti et al (2012) recognized the problem of looking at landslide occurrence and disregarding non-occurrence when applying the ID threshold. They used a Bayesian probability approach to derive the probabilistic transition explicitly taking into account landslide occurrence and non-occurrence. Also the role of hydrology in landslide initiation, although often acknowledged to be of key importance, is usually not included in the statistical precipitation ID threshold approach. Several attempts to more explicitly include hydrological predisposing factors have been proposed mainly including measures for antecedent soil moisture content (e.g. Crozier and Eyles, 1980; Glade et al, 2000; Godt et al, 2006; Ponziani et al, 2012) or by splitting data sets in physiographic units like lithology, soil type, land use or season (e.g. Sidle and Ochiai, 2006; Baum and Godt, 2010; Napolitano et al 2016; Peruccacci et al, 2017). These approaches improved the predictive accuracy of the ID thresholds. However, to the authors' knowledge, such studies have not been subject to a more thorough analysis of the specific hydrological information needed for reliable local and regional hazard prediction.

Therefore, the objectives of this invited perspective are to: (a) critically analyse the precipitation ID thresholds for shallow landslides and debris flows from a hydro-meteorological point of view; and (b) propose a conceptual framework for lumped *hydro-meteorological* hazard assessment based on the concepts of trigger and cause. We will frame in this perspective some published examples and associated discussions, making reference to work made by colleagues who have already explored this avenue. AimThe aim of this paper is to contribute to the development of a stronger conceptual model for regional landslide hazard assessment based on physical process understanding and. not only on empirical data.

## 2 HYDROMETEOROLOGICAL ANALYSIS OF ID THRESHOLDS

COMPARING ID THRESHOLDS WITH IDF CURVES

Both precipitation intensity-duration thresholds (ID) and precipitation intensity-duration-
frequency curves (IDF) are empirical relationships linking the duration of a precipitation
event, D, with its average intensity, $I = H/D$, H being the precipitation depth  during the
event. IDF curves are routinely used in stormwater and flood management design
and predictions, as they describe the relationship linking duration and mean intensity
of precipitation events characterized by the same return period (Chow et al., 1988). Several
functional expressions can be used to describe such a relationship (Bernard, 1932; Wenzel,
1982; Koutsoyiannis, 1998), most of which can be approximated, especially for durations
longer than 1 hr, as a power law:

$$I = A \times D^B \qquad (1)$$

with B [-] being the slope of the log-plotted straight line and A [L/T] a measure of the
rain intensity of a rain event of unit duration.

Equation (1) is also adopted to describe precipitation ID thresholds, the difference
being that the IDF curves are isolines of cumulative probability of precipitation events,
whereas the ID plots are empirical thresholds for shallow landslides and debris flow
occurrence. Figure 1 gives examples of IDF curves with a return period of 10 years from
different places around the world. A common feature of the curves is that, regardless of
geographic location, B ranges from -0.8 to -0.65 for rain durations longer than ~1 hour, while
it levels off to around -0.5 for $D \leq 1$ hr for most IDF curves. Note that IDF curves are mostly
determined for rain durations up to 24 hrs. In the same graph, the upper envelope of the
largest precipitation values ever observed (World Meteorological Organization, 1986), is
plotted using the equation proposed by Brutsaert (2005), which has a smaller slope with B
equal to -0.52.

[Figure]

Figure 1. Examples of intensity-duration-frequency curves for 10 year return period (1-9) and curve of the maximum observed precipitations (10). Location and source: 1 Najran region, Saudi Arabia (Elsebaie, 2012); 2

Uccle, Belgium (Van de Vyver, 2015); 3 Naples, Italy (Rossi and Villani, 1993); 4 Los Angeles, California (Wenzel, 1982); 5 Pelotas, Brazil (Damé et al., 2016); 7 Hamada, Japan (Iida, 2004)); 8 Selangor, Malaysia (Chang et al., 2015); 9 Sylhet, Bangladesh (Rasel and Hossain, 2015); 10 Greatest known observed point rainfall (Brutsaert, 2005).

[Figure]

[Figure]

Figure 2. Rainfall intensity-duration (ID) thresholds. Numbers refer to case studies (Guzzetti et al, 2007). Very
thick lines are global thresholds; thick lines are regional thresholds and thin lines are local thresholds. Black
lines show global thresholds and thresholds determined for regions or areas pertaining to the Central to Eastern
European region. Grey lines show thresholds determined for other regions or areas not pertaining to this area.

Figure 2 shows More than 90% of the thresholds that come from a global dataset of
landslides, more than 90% of which in the global data set are shallow landslides and debris
flows (Figure 2, Guzzetti et al, 2007). Note that the thresholds arethreshold is usually obtained
as lower envelope of the events resulting in landslide initiation, although also other
thresholdsthreshold definitions exist (e.g. Staley et al., 2013; Ciavolella et al, 2016; Peres and
Cancelliere, 2016, Ciavolella et al, 2016). Obviously, ID thresholds differ greatly between
climate and physiographic regions, especially in absolute values. Therefore, scaled
representations have been proposed for the thresholds, such as dividing precipitation intensity
by the mean annual precipitation (Guzzetti et al, 2007), in order to better compare the
thresholds. (Guzzetti et al, 2007). However, in our analysis the focus will beis on the
absoluteunscaled measured precipitation ID representation, as it is a convenient way to
compare with IDF and for the following discussion. The exponent of most of the reported
thresholds for initiation of landslides range between -0.2 and -0.6. By overlaying IDF and ID
curves (Figure 3), forFor landslides triggered by short precipitation events (D ≤ 1 hr), mostly
debris flows and some shallow landslides, the slopes of IDthe IDF and IDFID curves
substantially coincide. (Figure 3).. On the other hand, for longer precipitation durations, ID
thresholds have smaller slopes than IDF curves. This means that landslide initiation on the right side of the graph (lower precipitation intensity with longer duration) would occur with rapidly increasing return periods of precipitation events. This is counter-intuitive, as during a long-lasting wet period landslides are usually more frequent, while many debris-flows triggered by very short and intense storm originate from channel bed mobilization rather than being (new) mass movements. This shows that the method used to derive ID thresholds for landslide initiation based on landslide and precipitation reports leads to troublesome interpretations. Owing to the high spatial variability of rainfall at scales smaller than 5 km (e.g. Krajewski et al., 2003; Ciach and Krajewski, 2006) and the limited density of operational rain gauge networks, the rainfall intensity observed by rain gauges systematically underestimates the actual triggering rainfall intensity at debris flow initiation locations (Marra et al., 2016), especially for short rain duration and high return period (Destro et al., 2017). This issue has been shown to significantly affect the obtained ID thresholds for debris flow initiation (Nikolopoulos et al., 2015; Marra et al., 2017). Additionally, different methods adopted to define the dry period separating rain events have been shown to strongly affect the ID threshold (e.g. Vessia et al., 2014; Melillo et al; 2015). Furthermore, several authors already pointed out that characterizing a storm with its mean intensity, thus neglecting peaks and underestimating actual intensity, affects the estimated probability of landslide occurrence (e.g. D'Odorico et al., 2005; Peres and Cancelliere, 2016), and such an issue is obviously more significant for long storm durations. In fact, for the rainfall depth data used to derive IDF curves, the considered duration is simply a moving interval along the rainfall time series, regardless from the actual beginning and end of a rainfall event. So, the corresponding mean intensity usually refers, especially for short durations, to the heaviest part of a longer rainfall event. Conversely, whatever the criterion adopted for the definition of a rainfall event, in the case of ID threshold curves, the plotted mean intensity refers to the entire rainfall event. Thus, within very long events leading to landslide triggering, there is very likely an intensity peak, which is the "real" landslide trigger, preceded by a period of rain which contributes to predispose the slope to failure. In the (D, I) plane, the point corresponding to the peak would be shifted left- and upwards, compared to the point of the entire event. Given the typical slopes of IDF and ID threshold curves, this shift likely corresponds to a smaller return period.

[Figure]

Figure 3. Schematic representation of precipitation IDF curves, isolines of accumulated precipitation (ΣP) and  ID threshold for shallow landslides and debris flows (simplified from Figure 2).

HYDROLOGICAL INTERPRETATION OF ID THRESHOLDS

[revised manuscript text omitted]

Concerning the 'trigger'-axis, there is little debate; it is the rainfall intensity responsible for the short-term last push initiating a landslide. The time-scale for local and regional assessment of course depends on the local situation, but hourly or daily time-scales are the most common. The 'cause'-axis should represent the predisposing condition of the area under study. For hydrologically triggered landslides, it should be related to the antecedent wetness state of the area. However, there are several possible choices of hydrological variables to be plotted along the 'cause'-axis, such as (effective) soil water content, relative catchment storage and representative regional groundwater level.

The choice for the 'trigger' and 'cause' also implies a definition of the time-scale separating trigger from cause, which should be related to the characteristics of the triggered landslide, but is in practice often limited by the (temporal resolution of the) available data.

As mentioned before, there are – besides the soil moisture storage calculations previously described - various examples of hydrological information added to landslide thresholds. Hashino and Murota (1971) published  an analysis of landslide triggers in a catchment related to debris production using measured river discharge data to link the landslide triggers to the water balance of the catchment. They identified that the landslides in their study area occurred during above average antecedent conditions

. This is one of the earliest reported studies we know of explicitly looking at catchment water balance as an important source of information on the antecedent hydrological condition of an area in relation to landslide occurrence. Reichenbach et al (1998) made a combined flood and landslide hazard analysis of the Tiber river catchment using 72 years of historical daily discharge data from different gauging stations where hydrological parameters were calculated, such as maximum mean daily discharge, specific discharge  and flood volume and duration. Probability of occurrence of landslides and floods was based on the ranking of the events. Combining maximum mean daily discharge
and discharge intensity, regional hydrological thresholds for landslide and flood hazard
(individually or combined) could be drawn. Chitu et al (2016) followed a somewhat similar
approach, analyzing the river discharge in several catchments in the Ialomita Subcarpathians
in Romania for landslide events  in 2014. The catchments could be characterized as
having low/high relative storage. Additionally, a calibrated regional rainfall-runoff model was
used for hydrological analysis of landslides in specific catchments. Detailed analysis
of the (modelled) hydrological response indicated that in two catchments with low
infiltration capacity the direct runoff was strongly related to landslide occurrence,
whereas it could be linked to modeled soil infiltration flux in another
catchment. Extending the above to deep-seated landslides, the connected
regional groundwater level could be informative. Bogaard et al (2013) studied the hydro-
meteorological triggering threshold of the re-activating coastal Villerville–Cricqueboeuf
landslide, Normandy, France. In this situation the hinterland of the coastal cliff consists of a
well-defined regional groundwater level. Landslide reactivation was seen to take place only
when water level was in the upper, more permeable top layer. The triggering rain event
together with surpassing a certain regional groundwater threshold could explain 3 of the 4 re-
activations. Note that these groundwater levels were not taken in the active landslide area but
several kilometers inland.

Recently, further attempts have been made to use river discharge and lumped water
storage in a catchment as a proxy for the predisposing conditions for landslides along its
slopes. Following Hashino and Murota (1971), the basic idea is that when 'more-than-
average' water is stored in the catchment, it is more likely that a rainfall event triggers
landslides. The disadvantage of using catchment wide storage is the relatively low spatial
resolution, and the difficulty of having (reliable and homogeneous) discharge time series in
catchments. Moreover, catchment storage assessment needs information on evaporation which
can have significant uncertainties. Of course, such an approach works only if the causes of the
predisposing conditions for landslides are somewhat related to catchment scale hydrological
processes. Ciavolella et al. (2016) defined a cause-trigger hydro-meteorological threshold in
the catchment of river Scoltenna, in Emilia Romagna (Italy), linking catchment storage and
event rainfall intensity, and compared its performance with that of a statistical ID
precipitation threshold. The two thresholds performed similarly, with the hydro-
meteorological thresholds being more accurate for identifying
landslides, but giving a somewhat larger number of false positives.

These examples indicate that considering hydrological causes could be
useful for a better identification of landslide initiation, but, at the same time, they show that
the correct identification of the hydrological processes involved in the establishment of the
predisposing conditions for landslides  is mandatory for choosing the most
informative hydrological variable to be plotted along the x-'cause'-axis.

4 CONCLUDING REMARKS AND OUTLOOK

The intrinsic limitations of precipitation ID thresholds for the identification of
landslide initiation conditions has been noted for some time. Indeed,
such thresholds neglect the role of the hydrological processes occurring along slopes, which
predispose hillslopes to  failure (causes) and focus
predominately on the characteristics of the last rainfall events leading to slope failure
(triggers). As a consequence, the predictive accuracy of the ID thresholds is often low,
even when they refer to small areas. We argue that the threshold values for rainfall intensity
of short and long duration (the far left and right side of the graphs) have limited physical
meaning and, consequently, that the use of precipitation ID thresholds can lead to misleading
interpretations of initiation conditions, as important antecedent conditions and
rainfall intensity variations are not taken into account. For this reason, we here advocate to be
very careful in uncritically using the precipitation ID thresholds as kind of regional
characteristic of (shallow) landslide occurrence.

Equally, for this and several other reasons, many colleagues advocate the use of
spatially-distributed physically- based models for assessing landslide probability. The obvious
downside is that large data input and a well calibrated model are required. However, it is fair
to say, data are becoming more and more available and even precipitation predictions are
improving rapidly, especially with short lead-time. The use of high quality rainfall prediction
with very short lead time (e.g. 3 hours) requires efficient numerical models combined with
high computational power, especially if predictions are used for early warning purposes. This,
in practice, is still easier said than done. Therefore, we believe, that lumped, empirical (or
semi-empirical) thresholds will continue having a practical value, which still justifies
scientific attention.

We propose to use the cause-trigger concept for defining regional landslide initiation
thresholds. This, we agree, is challenging, but, in our opinion, not impossible. First of all, it is
needed to define the characteristic time-scale separating the (dynamic) long-term predisposing hydrological cause from the short 'final' landslide hazard triggering. This obviously depends on the landslide type and physiographic characteristics. Looking at the discussed examples, it becomes clear that the choice of the most informative hydrological variable to be used as a proxy for landslide predisposing conditions strictly depends on site-specific geomorphological characteristics, and that accurate analysis of the boundaries through which the potentially unstable area exchanges water with the surrounding hydrological systems is mandatory. In other words, for the assessment of landslide predisposing conditions, the water balance of the slope should be assessed, but getting the information about the inherent hydrological processes (e.g. evaporation, runoff, groundwater recharge) at the required spatial-temporal resolution is often a challenge and could require some kind of calculations or modelling. However, rapidly more and higher resolution hydrological data become available which can be used in assessing the hydrological predisposing condition.

ACKOWLEDGEMENT

This invited perspectives paper was motivated based on frequent discussions with many colleagues working in the field of landslide hazard assessment. We are grateful to them all. Also the excellent reviews by one anonymous reviewer, Roy Sidle, Francesco Marra and Ben Mirus and his colleagues from USGS are greatly acknowledged. We thank them for pointing to omissions and the many insightful comments which considerably improved the paper.